# NEURO-SYMBOLIC FORWARD REASONING

## ABSTRACT

Reasoning is an essential part of human intelligence and thus has been a long-standing goal in artificial intelligence research. With the recent success of deep learning, incorporating reasoning with deep learning systems, i.e., neuro-symbolic AI has become a major field of interest. We propose the Neuro-Symbolic Forward Reasoner (NSFR), a new approach for reasoning tasks taking advantage of differentiable forward-chaining using first-order logic. The key idea is to combine differentiable forward-chaining reasoning with object-centric (deep) learning. Differentiable forward-chaining reasoning computes logical entailments smoothly, i.e., it deduces new facts from given facts and rules in a differentiable manner. The object-centric learning approach factorizes raw inputs into representations in terms of objects. Thus, it allows us to provide a consistent framework to perform the forward-chaining inference from raw inputs. NSFR factorizes the raw inputs into the object-centric representations, converts them into probabilistic ground atoms, and finally performs differentiable forward-chaining inference using weighted rules for inference. Our comprehensive experimental evaluations on object-centric reasoning data sets, 2D *Kandinsky patterns* and 3D *CLEVR-Hans*, and a variety of tasks show the effectiveness and advantage of our approach.

## 1 INTRODUCTION

Right from the time of Aristotle, reasoning has been in the center of the study of human behavior (Miller, 1984). Reasoning can be defined as the process of deriving conclusions and predictions from available data. The long-lasting goal of artificial intelligence has been to develop rational agents akin to humans, and reasoning is considered to be a major part of achieving rationality (Johnson-Laird, 2010). Logic, both propositional and first-order, is an established framework to perform reasoning on machines (Boole, 1847). Such logical reasoning has been an essential part of the growth of machine learning over the years (Poole et al., 1987; Bottou, 2014; Dai et al., 2019) and has also given rise to statistical relational learning (Koller et al., 2007; Raedt et al., 2016) and probabilistic logic programming (Lukasiewicz, 1998; De Raedt & Kersting, 2003; De Raedt & Kimmig, 2015).

Object-centric reasoning has been widely addressed (Johnson et al., 2017; Mao et al., 2019; Chen et al., 2021; Han et al., 2019), where the task is to perform reasoning to answer the questions that are about the *objects* and its *attributes*. However, the task is challenging because the models should perform low-level visual perception and reasoning on high-level concepts. To mitigate this challenge, with the recent success of deep learning, incorporating logical reasoning with deep learning systems, i.e., neuro-symbolic AI has become a major field of interest (De Raedt et al., 2019; Garcez et al., 2019). It has the advantage of combining the expressivity of neural networks with the reasoning of symbolic methods.

Various benchmarks and methods have been developed for object-centric reasoning (Locatello et al., 2020b; Nanbo et al., 2020). Recently, data sets such as *Kandinsky patterns* (Mueller & Holzinger, 2019; Holzinger et al., 2019; 2021) and *CLEVR* (Johnson et al., 2017) have been proposed to assess the performance of the machine learning systems in object-centric reasoning tasks. For example, Figure 1 shows an example of the Kandinsky pattern: *"the figure has two pairs of objects with the same shape"* where Fig. (a) is following the pattern and Fig. (b) is not. Kandinsky patterns are inspired by human IQ-tests (Bruner et al., 1956; Dowe & Hernández-Orallo, 2012; Liu et al., 2019), which require humans to think on abstract patterns. The key feature of Kandinsky Patterns is its complexity, e.g., the arrangement of objects, closure or symmetry, and a group of objects.

Many approaches have been investigated for object-centric reasoning under the umbrella of neuro-symbolic learning (Rocktäschel & Riedel, 2017; Yang et al., 2017; Šourek et al., 2018; Manhaeve et al., 2018; Si et al., 2019; Mao et al., 2019; Cohen et al., 2020; Riegel et al., 2020). However, using these approaches it is difficult, if not impossible, to solve object-centric reasoning tasks such as Kandinsky patterns due to several underlying challenges: (i) the perception of the objects from the raw inputs and (ii) the reasoning on the attributes and the relations to capture the complex patterns (of varying size).

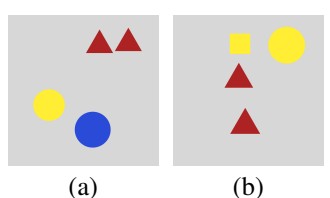

(a)          (b)

Figure 1: Examples of Kandinsky patterns. (a) follows the pattern: *"the figure has 2 pairs of objects with the same shape"*, but (b) isn't

In this work, we propose the **N**euro-**S**ymbolic **F**orward **R**easoner (NSFR), a novel neuro-symbolic learning framework for complex object-centric reasoning tasks. The key idea is to combine neural-based object-centric learning models with the differentiable implementation of first-order logic. It has *three* main components: (i) object-centric perception module, (ii) facts converter, and (iii) differentiable reasoning module. The object-centric perception module extracts information for each object and has been widely addressed in the computer vision community (Locatello et al., 2020a; Redmon et al., 2016). Facts converter converts the output of the visual perception module into the form of probabilistic logical atoms, which can be fed into the reasoning module. Finally, differentiable reasoning module performs the differentiable forward-chaining inference from a given input. It computes the set of ground atoms that can be deduced from the given set of ground atoms and weighted logical rules (Evans & Grefenstette, 2018; Shindo et al., 2021). The final prediction can be made based on the result of the forward-chaining inference.

Overall, we make the following contributions: (1) We propose Neuro-Symbolic Forward Reasoner (NSFR), a new neuro-symbolic learning framework that performs differentiable forward-chaining inference from visual data using object-centric models. NSFR can solve problems involving complex patterns on objects and attributes, such as the arrangement of objects, closure, or symmetry. (2) To establish NSFR, we show an extended implementation of the differentiable forward-chaining inference to overcome the scalability problem. Moreover, NSFR can take advantage of some essential features of the underlying neural network, such as batch computation, for logical reasoning. (3) To establish NSFR, we provide a conversion algorithm from object-centric representations to probabilistic facts. We propose *neural predicates*, which are associated with a function to produce a probability of a fact and yield a seamless combination of sub-symbolic and symbolic representations. (4) We empirically show that NSFR solves object-centric reasoning tasks more effectively than the SOTA logical and deep learning models. Furthermore, NSFR classifies complex patterns with high accuracy for 2D and 3D data sets, outperforming pure neural-based approaches for image recognition.

## 2 BACKGROUND AND RELATED WORK

**Notation.** We use bold lowercase letters $\mathbf{v}, \mathbf{w}, \ldots$ for vectors and the functions that return vectors. We use bold capital letters $\mathbf{X}, \ldots$ for tensors. We use calibrate letters $\mathcal{C}, \mathcal{A}, \ldots$ for (ordered) sets and typewriter font $\mathrm{p}(\mathrm{X}, \mathrm{Y})$ for terms and predicates in logical expressions (Appendix A.1 for details).

**Preliminaries.** We consider function-free first-order logic. *Language* $\mathcal{L}$ is a tuple $(\mathcal{P}, \mathcal{T}, \mathcal{V})$, where $\mathcal{P}$ is a set of predicates, $\mathcal{T}$ is a set of constants, and $\mathcal{V}$ is a set of variables. A *term* is a constant or a variable. We assume that each term has a *datatype*. A datatype $\mathrm{dt}$ specifies a set of constants $dom(\mathrm{dt}) = \mathcal{T}_{\mathrm{dt}} \subseteq \mathcal{T}$. We denote $n$-ary predicate $\mathrm{p}$ by $\mathrm{p}/(n, [\mathrm{dt}_1, \ldots, \mathrm{dt}_n])$, where $\mathrm{dt}_i$ is the datatype of $i$-th argument. An *atom* is a formula $\mathrm{p}(\mathrm{t}_1, \ldots, \mathrm{t}_n)$, where $\mathrm{p}$ is an $n$-ary predicate symbol and $\mathrm{t}_1, \ldots, \mathrm{t}_n$ are terms. A *ground atom* or simply a *fact* is an atom with no variables. A *literal* is an atom or its negation. A *positive literal* is just an atom. A *negative literal* is the negation of an atom. A *clause* is a finite disjunction ($\lor$) of literals. A *definite clause* is a clause with exactly one positive literal. If $A, B_1, \ldots, B_n$ are atoms, then $A \lor \neg B_1 \lor \ldots \lor \neg B_n$ is a definite clause. We write definite clauses in the form of $A : -B_1, \ldots, B_n$. Atom $A$ is called the *head*, and set of negative atoms $\{B_1, \ldots, B_n\}$ is called the *body*. We denote special constant *true* as $\top$ and *false* as $\bot$. Substitution $\theta = \{\mathrm{X}_1 = \mathrm{t}_1, ..., \mathrm{X}_n = \mathrm{t}_n\}$ is an assignment of term $\mathrm{t}_i$ to variable $\mathrm{X}_i$. An application of substitution $\theta$ to atom $A$ is written as $A\theta$.

**Related Work.** Reasoning with neuro-symbolic systems has been studied extensively for various applications such as ocean study (Corchado, 1995), business internal control (Corchado et al., 2004) and forecasting (Fdez-Riverola et al., 2002). More recently, several neuro-symbolic techniques for commonsense reasoning (Arabshahi et al., 2021), visual question answering (Mao et al., 2019; Amizadeh et al., 2020) and multimedia tasks (Khan & Curry, 2020) have been developed. They either do not employ a differentiable forward reasoner or miss objet-centric learning in the end-to-end reasoning architecture.

Object-centric learning is an approach to decompose an input image into representations in terms of objects (Dittadi et al., 2021). This problem has been widely addressed in the computer vision community. The typical approach is the object detection (or supervised) approach such as Faster-RCNN (Ren et al., 2015) and YOLO (Redmon et al., 2016). Another approach is the unsupervised approach (Burgess et al., 2019; Engelcke et al., 2020; Locatello et al., 2020a), where the models acquire the ability of object-perception without or fewer annotations. These two different paradigms have different advantages. NSFR encapsulates different object-perception models, thus allows us to choose a proper model depending on the situation and the problem to be solved.

Also, the integration of symbolic logic and neural networks has been addressed, see e.g. Deep-Problog (Manhaeve et al., 2018) and NeurASP (Yang et al., 2020). The key difference from the past approaches is that NSFR supports essential features of neural networks such as batch computation and that it is fully differentiable. Thus it scales well to large data sets, leading to several avenues for future work learning neural networks with logical constraints (Hu et al., 2016; Xu et al., 2018).

## 3 THE NEURO-SYMBOLIC FORWARD REASONER (NSFR)

Let us now introduce the Neuro-Symbolic Forward Reasoner (NSFR) in four steps. First, we give an overview of the problem setting and the framework. Second, we specify a language of first-order logic focusing on the object-centric reasoning tasks. Third, we explain the facts converter. Finally, we show the differentiable forward-chaining inference algorithm, an extended implementation from the previous approaches.

### 3.1 OVERVIEW

**Problem Scenario.** We address the image classification problem, where each image contains objects, and the classification rules are defined on the relations of objects and their attributes. We define the *object-centric reasoning problem* as follows:

**Definition 1** *An **Object-Centric Reasoning Problem** $\mathcal{Q}$ is a tuple $(\mathcal{I}^+, \mathcal{I}^-, P)$, where $\mathcal{I}^+$ is a set of images that follow pattern $P$, $\mathcal{I}^-$ is a set of images that do not follow pattern $P$. Each image $X \in \mathcal{I}^+ \cup \mathcal{I}^-$ contains several objcets, and each object has its attributes. Pattern $P$ is a pattern that is described as logical rules or natural language sentence, which is defined on the attributes and relations of objects. The solution of problem $\mathcal{Q}$ is a set of binary labels $\mathcal{Y} = \{y_i\}_{i=0,...,N}$ for each image $X_i \in \mathcal{I}^+ \cup \mathcal{I}^-$, where $N = |\mathcal{I}^+ \cup \mathcal{I}^-|$.*

**Architecture Overview.** NSFR performs object-centric perception from raw input and reasoning on the extracted high-level concepts. Figure 2 presents an overview of NSFR. First, NSFR perceives objects from raw input whose output is a set of vectors, called object-centric representations, where each vector represents each object in the input. Then, the fact converter takes these object-centric representations as input and returns a set of probabilistic facts. The probabilistic facts are then fed into the reasoning module, which performs differentiable forward-chaining inference using weighted rules. Finally, the final prediction is made on the result of the inference. We briefly summarize the steps of the process as follows:

**Step 1:** Let $\mathbf{X} \in \mathbb{R}^{B \times N}$ be a batch of input images. Perception function $f_{percept} : \mathbb{R}^{B \times N} \to \mathbb{R}^{B \times E \times D}$ factorizes input $\mathbf{X}$ into a set of object-centric representations $\mathbf{Z} \in \mathbb{R}^{B \times E \times D}$, where $E \in \mathbb{N}$ is the number of objects, $D \in \mathbb{N}$ is the dimension of object-centric vector.

**Step 2:** Let $\mathcal{G}$ be a set of ground atoms. Convert function $f_{convert} : \mathbb{R}^{B \times E \times D} \times \mathcal{G} \to \mathbb{R}^{B \times G}$ generates a probabilistic vector representation of facts, where $G = |\mathcal{G}|$.

**Step 3:** Infer function $f_{infer} : \mathbb{R}^{B \times G} \to \mathbb{R}^{B \times G}$ computes forward-chaining inference using weighted clauses $\mathcal{C}$.

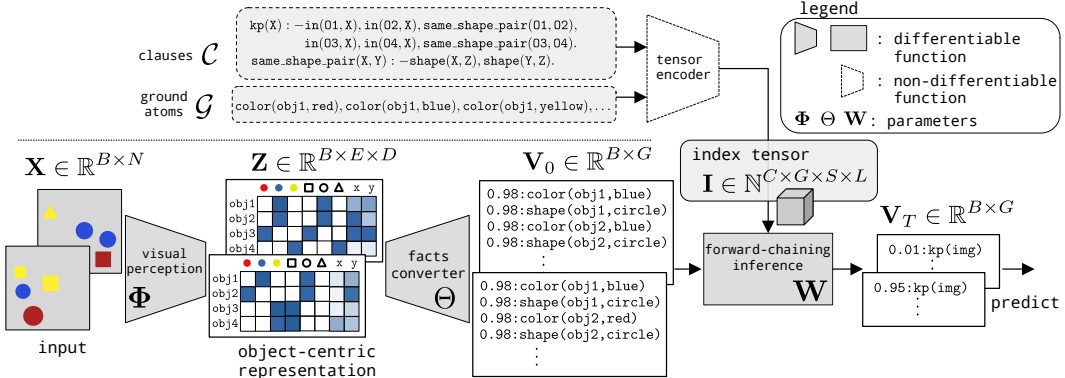

Figure 2: An overview of NSFR. The object-centric model produces outputs in terms of objects. The facts converter obtains probabilistic facts from the object-centric representation. The differentiable forward-chaining inference computes the logical entailment softly from the probabilistic facts and weighted rules. The final prediction is computed based on the entailed facts.

**Step 4:** Predict function $f_{predict} : \mathbb{R}^{B \times G} \to \mathbb{R}^B$ computes the probability of target facts. The probability of the labels $\mathbf{y}$ of the batch of input $\mathbf{X}$ is computed as:

$$p(\mathbf{y}|\mathbf{X}) = f_{pred}(f_{infer}(f_{convert}(f_{percept}(\mathbf{X}; \mathbf{\Phi}), \mathcal{G}; \mathbf{\Theta}); \mathcal{C}, \mathbf{W})), \tag{1}$$

where $\mathbf{\Phi}, \mathbf{\Theta}$, and $\mathbf{W}$ are learnable parameters.

We now present each component of our architecture in detail.

## 3.2 OBJECT-CENTRIC REASONING LANGUAGE

We have to define a first-order logic language to build a consistent neuro-symbolic framework for object-centric reasoning. Intuitively, we assume that all constants are divided into *inputs*, *objects*, and *attributes*, and the attribute constants have different data types such as *colors* and *shapes*.

**Definition 2** An **Object-Centric Reasoning Language** is a function-free language $\mathcal{L} = (\mathcal{P}, \mathcal{T}, \mathcal{V})$, where $\mathcal{P}$ is a set of predicates, $\mathcal{T}$ is a set of constants, and $\mathcal{V}$ is a set of variables. The set of constants $\mathcal{T}$ is divided to a set of inputs $\mathcal{X}$, a set of objects $\mathcal{O}$, and a set of attributes $\mathcal{A}$, i.e., $\mathcal{T} = \mathcal{X} \cup \mathcal{O} \cup \mathcal{A}$. The attribute constants $\mathcal{A}$ is devided into a set of constants for each datatype, i.e., $\mathcal{A} = \mathcal{A}_{\mathtt{dt_1}} \cup \ldots \cup \mathcal{A}_{\mathtt{dt_n}}$, where $\mathcal{A}_{\mathtt{dt_i}}$ is a set of constants of the $i$-th datatype $\mathtt{dt_i}$.

**Example 1**: The language for Figure 2 can be represented as $\mathcal{L}_1 = (\mathcal{P}, \mathcal{T}, \mathcal{V})$, where $\mathcal{P} = \{\mathtt{kp}/(1, [\mathtt{image}]), \mathtt{in}/(2, [\mathtt{object}, \mathtt{image}]), \mathtt{color}/(2, [\mathtt{object}, \mathtt{color}]), \mathtt{shape}/(2, [\mathtt{object}, \mathtt{shape}]), \mathtt{same\_shape\_pair}(2, [\mathtt{object}, \mathtt{object}])\}$, and $\mathcal{T} = \mathcal{X} \cup \mathcal{O} \cup \mathcal{A}_{\mathtt{color}} \cup \mathcal{A}_{\mathtt{shape}}$ where $\mathcal{X} = \{\mathtt{img}\}, \mathcal{O} = \{\mathtt{obj1}, \mathtt{obj2}, \mathtt{obj3}, \mathtt{obj4}\}, \mathcal{A}_{\mathtt{color}} = \{\mathtt{red}, \mathtt{yellow}, \mathtt{blue}\}$ where $\mathcal{A}_{\mathtt{shape}} = \{\mathtt{square}, \mathtt{ciecle}, \mathtt{triangle}\}$, and $\mathcal{V} = \{\mathtt{X}, \mathtt{Y}, \mathtt{Z}, \mathtt{O1}, \mathtt{O2}, \mathtt{O3}, \mathtt{O4}\}$.

## 3.3 OBJECT-CENTRIC PERCEPTION

We make the minimum assumption that the perception function takes an image and returns a set of object-centric vectors, where each of the vectors represents each object. For simplicity, we assume that each dimension of the vector represents the probability of the attributes for each object. For example, suppose each object has *color, shape, position* as attributes. The color varies *red, blue, yellow*, the shape varies *square, circle, triangle*, and position is represented as a $(x, y)$-coordinates. In this case, each object can be represented as an 8-dim vector, as illustrated in Figure 2.

Let $N$ be the input size, $E$ be the maximum number of objects that can appear in one image, and $D$ be the number of attributes for each object. For a batch of input images $\mathbf{X} \in \mathbb{R}^{B \times N}$, the object-centric perception function $f_{percept} : \mathbb{R}^{B \times N} \to \mathbb{R}^{B \times E \times D}$ parameterized $\mathbf{\Phi}$ produces a batch of object-centric representations $\mathbf{Z} \in \mathbb{R}^{B \times E \times D}$: $\mathbf{Z} = f_{percept}(\mathbf{X}; \mathbf{\Phi})$. We note that each value $\mathbf{Z}_{i,j,k}$ represents the probability of the $k$-th attribute on the $j$-th object in the $i$-th image in the batch. We denote the tensor for $j$-th object as $\mathbf{Z}^{(j)} = \mathbf{Z}_{:,j,:} \in \mathbb{R}^{B \times D}$.

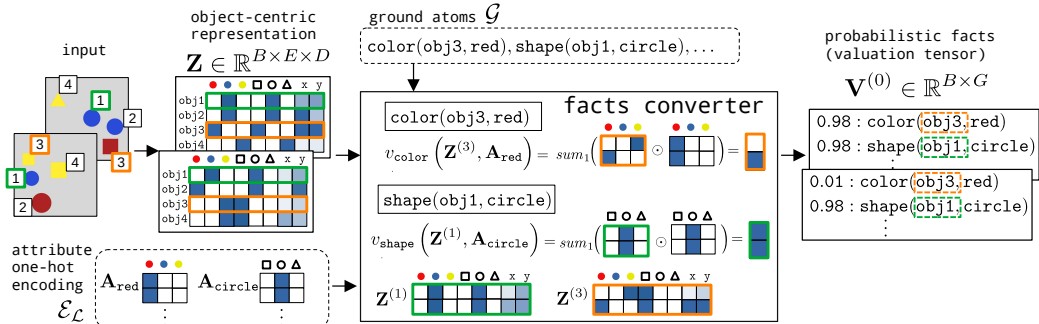

Figure 3: An overview of the facts-converting process. NSFR decomposes the raw-input images into the object-centric representations (left). The valuation functions are called to compute the probability of ground atoms (middle). The result is converted into the form of vector representations of the probabilistic ground atoms (right).

## 3.4 FACTS CONVERTER

After the object-centric perception, NSFR obtains the logical representation, i.e., a set of probabilistic ground atoms. We propose a new type of predicate that can refer to differentiable functions to compute the probability and a seamless converting algorithm from the perception result to probabilistic ground atoms.

### 3.4.1 TENSOR REPRESENTATIONS OF CONSTANTS

Specifically, in NSFR, constants are mapped to tensors as described below.

**Objects.** We map the object constants to the object-centric representation from the visual-perception module. The output of the visual-perception module is already factorized in terms of objects. Therefore the tensor for each object is extracted easily by slicing the output.

**Attributes.** We map the attribute constants to their corresponding one-hot encoding and assume that it is expanded to the batch size. Let $\mathcal{E}_{\mathcal{L}}$ be the set of one-hot encoding of attribute constants in language $\mathcal{L}$. For e.g., for language $\mathcal{L}_1$ in Example 1, color red has tensor representation as $\mathbf{A}_{\mathtt{red}} = [[1, 0, 0], [1, 0, 0]] \in \mathbb{R}^{2 \times 3}$, where the batch size is 2. We assume that we have the encoding for each attribute, i.e., $\mathcal{E}_{\mathcal{L}_1} = \{\mathbf{A}_{\mathtt{red}}, \mathbf{A}_{\mathtt{yellow}}, \mathbf{A}_{\mathtt{blue}}, \mathbf{A}_{\mathtt{square}}, \mathbf{A}_{\mathtt{circle}}, \mathbf{A}_{\mathtt{triangle}}\}$.

In summary, we define the tensor representations for each *object* and *attribute* constant t as:

$$f_{to\_tensor}(\mathtt{t}; \mathbf{Z}, \mathcal{E}_{\mathcal{L}}) = \begin{cases} \mathbf{Z}^{(i)} & \text{if } \mathtt{t} = \mathtt{obj_i} \in \mathcal{O} \\ \mathbf{A}_{\mathtt{dt}}^{(i)} & \text{if } \mathtt{t} = \mathtt{attr_i} \in \mathcal{A}_{\mathtt{dt}} \end{cases}, \tag{2}$$

where $\mathbf{A}_{\mathtt{dt}}^{(i)} \in \mathcal{E}_{\mathcal{L}}$ is the one-hot encoding of the $i$-th attribute of datatype dt. For example, $f_{to\_tensor}(\mathtt{obj1}; \mathbf{Z}, \mathcal{E}_{\mathcal{L}_1}) = \mathbf{Z}^{(1)}$ and $f_{to\_tensor}(\mathtt{red}; \mathbf{Z}, \mathcal{E}_{\mathcal{L}_1}) = \mathbf{A}_{\mathtt{red}} = [[1, 0, 0], [1, 0, 0]]$.

### 3.4.2 NEURAL PREDICATE

To solve the object-centric reasoning tasks, the model should capture the relation that is characterized by continuous features, for e.g., the *close by* relation between two objects. To encode such concepts into the form of logical facts, we introduce **_neural predicate_** that composes a ground atom associated with a differentiable function. Neural predicates computes the probability of the ground atoms using the object-centric representations from the visual perception module.

**Definition 3** *A neural predicate* $\mathtt{p}/(n, [\mathtt{dt_1}, \dots, \mathtt{dt_n}])$ *is a n-ary predicate associated with a function* $v_{\mathtt{p}} : \mathbb{R}^{d_1 \times \dots \times d_n} \to \mathbb{R}^B$, *where* $\mathtt{dt_i}$ *is the datatype of the i-th argument, and* $d_i \in \mathbb{N}$ *is the dimension of the tensor representation of the constant whose datatype is* $\mathtt{dt_i}$

**Example 2**: Figure 3 illustrates how the neural predicates and the valuation functions are computed. (1) For neural predicate $\mathtt{color}/(2, [\mathtt{object}, \mathtt{color}])$, the probability of ground atom $\mathtt{color}(\mathtt{obj3}, \mathtt{red})$ is computed by valuation function $v_{\mathtt{color}} : \mathbb{R}^{2 \times 5} \times \mathbb{R}^{2 \times 3} \to \mathbb{R}^2$ as

$v_{\texttt{color}}(\mathbf{Z}^{(3)}, \mathbf{A}_{\texttt{red}}) = sum_1(\mathbf{Z}^{(3)}_{:,0:3} \odot \mathbf{A}_{\texttt{red}})$, where $\mathbf{A}_{\texttt{red}} \in \{0,1\}^{2 \times 3}$ is a one-hot encoding of the color of $\texttt{red}$ that is expanded to the batch size, $sum_1$ is the sum operation for the dimension 1, and $\odot$ is the element-wise multiplication. (2) Likewise, for neural predicate $\texttt{shape}/(2, [\texttt{object}, \texttt{shape}])$, the probability of ground atom $\texttt{shape}(\texttt{obj1}, \texttt{circle})$ is computed by valuation function $v_{\texttt{shape}}$ : $\mathbb{R}^{2 \times 5} \times \mathbb{R}^{2 \times 3} \to \mathbb{R}^2$ as $v_{\texttt{shape}}(\mathbf{Z}^{(1)}, \mathbf{A}_{\texttt{circle}}) = sum_1(\mathbf{Z}^{(1)}_{:,3:6} \odot \mathbf{A}_{\texttt{circle}})$. (3) For neural predicate $\texttt{closeby}(2/[\texttt{object}, \texttt{object}])$, the probability of ground atom $\texttt{closeby}(\texttt{obj1}, \texttt{obj2})$ is computed by valuation functioin $v_{\texttt{closeby}}$ : $\mathbb{R}^{2 \times 5} \times \mathbb{R}^{2 \times 5} \to \mathbb{R}$ as: $v_{\texttt{closeby}}(\mathbf{Z}^{(1)}, \mathbf{Z}^{(2)}) = \sigma\left(norm_0\left(\mathbf{Z}^{(1)}_{:,4:6} - \mathbf{Z}^{(2)}_{:,4:6}\right); \mathbf{w}\right)$, where $norm_0$ is the norm function along dimension 0, $\sigma$ is the sigmoid function for each element of the input, and $\mathbf{w}$ is the trainable parameter. By adapting the parameters in neural predicates, NSFR can learn the concepts determined by numerical attributes and their relations. We note that valuation functions of neural predicates can be replaced by other differentiable functions, e.g., multilayer perceptrons.

### 3.4.3 Conversion Algorithm to Valuation Tensors

The facts converter produces a set of probabilistic ground atoms that are fed into the reasoning module. In NSFR, the probabilistic facts are represented in the form of tensors called *valuation tensors*.

**Valuation.** Valuation tensor $\mathbf{V}^{(t)} \in \mathbb{R}^{B \times G}$ maps each ground atom into a continuous value at each time step $t$. Each value $\mathbf{V}^{(t)}_{i,j}$ represents the probability of ground atom $F_j \in \mathcal{G}$ for the $i$-th example in the batch. The output of the perception module $\mathbf{Z} \in \mathbb{R}^{B \times E \times D}$ is compiled into initial valuation tensor $\mathbf{V}^{(0)}$. The differentiable inference function is performed based on valuation tensors. To compute the $T$-step forward-chaining inference, we compute the sequence of valuation tensors $\mathbf{V}^{(0)}, \ldots, \mathbf{V}^{(T)}$.

**Conversion into Valuation Tensors.** Neural predicates yield a seamless conversion algorithm from the object-centric vectors into the probabilistic facts. Algorithm 1 (see Appendix B) describes the converting procedure. For each ground atom, if it consists of a neural predicate, then the valuation function is called to compute the probability of the atom. We note that the valuation function computes probability in batch. NSFR allows background knowledge as a set of ground atoms. The probability of background knowledge is set to $1.0$.

### 3.5 Differentiable Forward-chaining Inference

NSFR performs reasoning based on the differentiable forward-chaining inference approach (Evans & Grefenstette, 2018; Shindo et al., 2021). The key idea is to implement the forward reasoning of first-order logic using tensors and operations between them using the following steps: **(Step 1)** Tensor $\mathbf{I}$, which is called *index tensor*, is built from given set of clauses $\mathcal{C}$ and fixed set of ground atoms $\mathcal{G}$. It holds the relationships between clauses $\mathcal{C}$ and ground atoms $\mathcal{G}$. Its dimension is proportional to $|\mathcal{C}|$ and $|\mathcal{G}|$. **(Step 2)** A computational graph is constructed from $\mathbf{I}$ and clause weights $\mathbf{W}$. The weights define probability distributions over clauses $\mathcal{C}$, approximating a logic program softly. The probabilistic forward-chaining inference is performed by the forwarding algorithm on the computational graph with input $\mathbf{V}^{(0)}$, which is the output of the fact converter. We now step-wise describe the process.

### 3.5.1 Tensor Encoding

We build a tensor that holds the relationships between clauses $\mathcal{C}$ and ground atoms $\mathcal{G}$. We assume that $\mathcal{C}$ and $\mathcal{G}$ are an ordered set, i.e., where every element has its own index. Let $L$ be the maximum body length in $\mathcal{C}$, $S$ be the maximum number of substitutions for existentially quantified variables in clauses $\mathcal{C}$, $C = |\mathcal{C}|$, and $G = |\mathcal{G}|$. Index tensor $\mathbf{I} \in \mathbb{N}^{C \times G \times S \times L}$ contains the indices of the ground atoms to compute forward inferences. Intuitively, $\mathbf{I}_{i,j,k,l}$ is the index of the $l$-th ground atom (subgoal) in the body of the $i$-th clause to derive the $j$-th ground atom with the $k$-th substitution for existentially quantified variables.

**Example 3**: Let $R_0 = \texttt{kp(X)} : -\texttt{in(O1, X)}, \texttt{shape(O1, square)} \in \mathcal{C}$ and $F_2 = \texttt{kp(img)} \in \mathcal{G}$, and we assume that object constants are $\{\texttt{obj1}, \texttt{obj2}\}$. To deduce fact $F_2$ using clause $R_0$,

$F_2$ and the head atom can be unified by substitution $\theta = \{\text{X} = \text{img}\}$. By applying $\theta$ to body atoms, we get clause $\text{kp}(\text{img}) : -\text{in}(\text{O1}, \text{img}), \text{shape}(\text{O1}, \text{square}).$, which has an existentially quantified variable $\text{O1}$. By considering the possible substituions for $\text{O1}$, we have grounded clauses as $\text{kp}(\text{img}) : -\text{in}(\text{obj1}, \text{img}), \text{shape}(\text{obj1}, \text{square}), \text{kp}(\text{img}) : -\text{in}(\text{obj2}, \text{img}), \text{shape}(\text{obj2}, \text{square})$. Then the following table shows tensor $\mathbf{I}_{0,:,0,:}$ and $\mathbf{I}_{0,:,1,:}$:

| $j$ | 0 | 1 | 2 | 3 | 4 | 5 | ... |
|---|---|---|---|---|---|---|---|
| $\mathcal{G}$ | $\perp$ | $\top$ | $\text{kp}(\text{img})$ | $\text{in}(\text{obj1}, \text{img})$ | $\text{in}(\text{obj2}, \text{img})$ | $\text{shape}(\text{obj1}, \text{square})$ | ... |
| $\mathbf{I}_{0,j,0,:}$ | $[0,0]$ | $[1,1]$ | $[3,5]$ | $[0,0]$ | $[0,0]$ | $[0,0]$ | ... |
| $\mathbf{I}_{0,j,1,:}$ | $[0,0]$ | $[1,1]$ | $[4,6]$ | $[0,0]$ | $[0,0]$ | $[0,0]$ | ... |

Ground atoms $\mathcal{G}$ and the indices are represented on the upper rows in the table. For example, $\mathbf{I}_{0,2,0,:} = [3,5]$ because $R_0$ entails $\text{kp}(\text{img})$ with substitution $\theta = \{\text{O1} = \text{obj1}\}$. Then the subgoal atoms are $\{\text{in}(\text{obj1}, \text{img1}), \text{shape}(\text{obj1}, \text{square})\}$, which have indices $[3,5]$, respectively. With another substitution $\theta = \{\text{O1} = \text{obj2}\}$, the subgoal atoms are $\{\text{in}(\text{obj2}, \text{img1}), \text{shape}(\text{obj1}, \text{square})\}$, which have indices $[4,6]$, respectively. The atoms which have a different predicate, e.g., $\text{shape}(\text{obj1}, \text{square})$, will never be entailed by clause $R_0$. Therefore, the corresponding values are filled with 0, which represents the index of the *false* atom.

### 3.5.2 DIFFERENTIABLE INFERENCE

Using the encoded index tensor, NSFR performs differentiable forward-chaining reasoning. We briefly summarize the steps as follows. **(Step 1):** Each clause is compiled into a function that performs forward reasoning. **(Step 2):** The weighted sum of the results from each clause is computed. **(Step 3):** $T$-time step inference is computed by amalgamating the inference results recursively. We extend previous approaches (Evans & Grefenstette, 2018; Shindo et al., 2021) for batch computation.

**Clause Function.** Each clause $R_i \in \mathcal{C}$ is compiled in to a clause function. The clause function takes valuation tensor $\mathbf{V}^{(t)}$, and returns valuation tensor $\mathbf{C}_i^{(t)} \mathbb{R}^{B \times G}$, which is the result of 1-step forward reasoning using $R_i$ and $\mathbf{V}^{(t)}$. The clause function is computed as follows. First, tensor $\mathbf{I}_i \in \mathbb{R}^{G \times S \times L}$ is extended for batches, i.e., $\tilde{\mathbf{I}}_i \in \mathbb{N}^{B \times G \times S \times L}$, and $\mathbf{V} \in \mathbb{R}^{B \times G}$ is extended to the same shape, i.e., $\tilde{\mathbf{V}} \in \mathbb{R}^{B \times G \times S \times L}$. Using these tensors, the clause function is computed as:

$$\mathbf{C}_i^{(t)} = softor_3^\gamma(prod_2(gather_1(\tilde{\mathbf{V}}, \tilde{\mathbf{I}})), \tag{3}$$

where $gather_1(\mathbf{X}, \mathbf{Y})_{i,j,k,l} = \mathbf{X}_{i, \mathbf{Y}_{i,j,k,l}, k, l}$, and $prod_2$ returns the product along dimension 2. $softor_d^\gamma$ is a function for taking logical *or* softly along dimension $d$:

$$softor_d^\gamma(\mathbf{X}) = \frac{1}{S} \gamma \log\left(sum_d \exp\left(\mathbf{X}/\gamma\right)\right), \tag{4}$$

where $\gamma > 0$ is a smooth parameter, $sum_d$ is the sum function for tensors along dimension $d$, and

$$S = \begin{cases} 1.0 & \text{if } max\left(\gamma \log sum_d \exp\left(\mathbf{X}/\gamma\right)\right) \leq 1.0 \\ max\left(\gamma \log sum_d \exp\left(\mathbf{X}/\gamma\right)\right) & \text{otherwise} \end{cases}. \tag{5}$$

Normalization term $S$ ensures that the function returns the normalized probabilistic values. We refer appendix I for more details about the $softor_d^\gamma$ function. In Eq. 3, applying the $softor_d^\gamma$ function corresponds to considering all possible substitutions for existentially quantified variables in the body atoms of the clause and taking *logical or* softly over the results of possible substitutions. The results from each clause is stacked into tensor $\mathbf{C}^{(t)} \in \mathbb{R}^{C \times B \times G}$, i.e., $\mathbf{C}^{(t)} = stack_0(\mathbf{C}_1^{(t)}, \ldots, \mathbf{C}_C^{(t)})$, where $stack_0$ is a stack function for tensors along dimension 0.

**Soft (Logic) Program Composition.** In NSFR, a logic program is represented smoothly as a weighted sum of the clause functions following (Shindo et al., 2021). Intuitively, NSFR has $M$ distinct weights for each clauses, i.e., $\mathbf{W} \in \mathbb{R}^{M \times C}$. By taking softmax of $\mathbf{W}$ along dimension 1, $M$ clauses are softly chosen from $C$ clauses. The weighted sum of clause functions are computed as follows. First, we take the softmax of the clause weights $\mathbf{W} \in \mathbb{R}^{M \times C}$: $\mathbf{W}^* = softmax_1(\mathbf{W})$ where $softmax_1$ is a softmax function over the dimension 1. The clause weights $\mathbf{W}^* \in \mathbb{R}^{M \times C}$ and the output of the clause function $\mathbf{C}^{(t)} \in \mathbb{R}^{C \times B \times G}$ are expanded to the same shape $\tilde{\mathbf{W}}^*, \tilde{\mathbf{C}}^{(t)} \in$

| | Training Data | | | Test Data | | |
|---|---|---|---|---|---|---|
| | NSFR | ResNet50 | YOLO+MLP | NSFR | ResNet50 | YOLO+MLP |
| Twopairs | **1.0** | **1.0** | **1.0** | **1.0** | 0.50 | 0.98 |
| Threepairs | **1.0** | **1.0** | **1.0** | **1.0** | 0.515 | 0.912 |
| Closeby | **1.0** | **1.0** | **1.0** | **1.0** | 0.54 | 0.91 |
| Red-Triangle | 0.958 | **1.0** | **1.0** | **0.956** | 0.57 | 0.79 |
| Online/Pair | 0.997 | **1.0** | **1.0** | **1.0** | 0.52 | 0.66 |
| 9-Circles | 0.964 | **1.0** | **1.0** | **0.952** | 0.50 | 0.50 |

Table 1: The classification accuracy in each data set. NSFR outperforms the considered baselines. Neural networks over-fit while training and perform poorly with testing data. Best results are bold.

$\mathbb{R}^{M \times C \times B \times G}$. Then we compute tensor $\mathbf{H}^{(t)} \in \mathbb{R}^{M \times B \times G}$: $\mathbf{H}^{(t)} = sum_1(\tilde{\mathbf{W}}^* \odot \tilde{\mathbf{C}})$, where $\odot$ is element-wise multiplication, and $sum_1$ is a summation along dimension 1. Each value $\mathbf{H}_{i,j,k}^{(t)}$ represents the probability of $k$-th ground atom using $i$-th clause weights for the $j$-th example in the batch. Finally, we compute tensor $\mathbf{R}^{(t)} \in \mathbb{R}^{B \times G}$ corresponding to the fact that logic program is a set of clauses: $\mathbf{R}^{(t)} = softor_0^\gamma(\mathbf{H})$.

**Multi-step Forward-Chaining Reasoning.** We define the 1-step forward-chaining reasoning function as: $r(\mathbf{V}^{(t)}; \mathbf{I}, \mathbf{W}) = \mathbf{R}^{(t)}$ and compute the $T$-step reasoning by: $\mathbf{V}^{(t+1)} = softor_1^\gamma(stack_1(\mathbf{V}^{(t)}, r(\mathbf{V}^{(t)}; \mathbf{I}, \mathbf{W})))$, where $\mathbf{I} \in \mathbb{N}^{C \times G \times S \times L}$ is a precomputed index tensor, and $\mathbf{W} \in \mathbb{R}^{M \times C}$ is clause weights.

## 4 EXPERIMENTAL EVALUATION

We empirically demonstrate the following desired properties of NSFR on 2 data sets (see App. C): (i) NSFR solves object-centric reasoning tasks with complex abstract patterns, (ii) NSFR can handle complex 3d scenes, and (iii) NSFR can perform fast reasoning with the batch computation.

### 4.1 SOLVING KANDINSKY PATTERNS

**Data.** We adopted Kandinsky pattern data sets (Mueller & Holzinger, 2019; Holzinger et al., 2019; 2021), a relatively new benchmark for object-centric reasoning tasks and use 6 Kandinsky patterns.

**Model.** We used YOLO (Redmon et al., 2016) as a perception module and trained it on the pattern-free figures, which are randomly generated. The correct rules are given to classify the figures, and clause weights are initialized to choose each of them. For e.g., the classification rule of the *twopairs* data set is : *"the Kandinsky Figure has two pairs of objects with the same shape, in one pair the objects have the same colors in the other pair different colors, two pairs are always disjunct, i.e. they don't share objects."*. This can be represented as a logic program containing *four* clauses:

```
1 kp(X):-in(O1,X),in(O2,X),in(O3,X),in(O4,X),same_shape_pair(O1,O2),
      same_color_pair(O1,O2),same_shape_pair(O3,O4),diff_color_pair(O3,O4).
2 same_shape_pair(X,Y):-shape(X,Z),shape(Y,Z).
3 same_color_pair(X,Y):-color(X,Z),color(Y,Z).
4 diff_color_pair(X,Y):-color(X,Z),color(Y,W),diff_color(Z,W).
```

**Pre-training.** We generated 15k pattern-free figures for pre-training of the visual perception module. Each object has the class label and the bounding box as an annotation. We generated 5k concept examples for neural predicate `closeby` and `online`.

**Baselines.** We adopted ResNet (He et al., 2016) as a benchmark and also compare against YOLO+MLP, where the input figure is fed to the pre-trained YOLO model, and a simple MLP module predicts the class label from the YOLO outputs.

**Results.** Table 1 shows the results for each Kandinsky data set. The Resnet50 model overfits while training and thus performs poorly in every test data. The YOLO+MLP model performs comparatively better and achieves greater than 90% accuracy in *twopairs, threepairs*, and *closeby* data set. However, in relatively complex patterns of *red-traignle, online/pair*, and *9-cicle* data sets the performance degrades. On the contrary, NSFR outperforms the considered baselines significantly and achieves perfect classification in 4 out of the 6 data sets.

| Model | Validation | Test | Validation | Test |
|---|---|---|---|---|
| | CLEVR-Hans3 | | CLEVR-Hans7 | |
| CNN | 99.55 | 70.34 | 96.09 | 84.50 |
| NeSy (Default) | 98.55 | 81.71 | 96.88 | 90.97 |
| NeSy-XIL | 100.00 | 91.31 | 98.76 | **94.96** |
| NS-FR | 98.18 | **98.40** | 93.60 | 92.19 |

Table 2: Classification accuracy for CLEVR-Hans data sets compared to baselines.

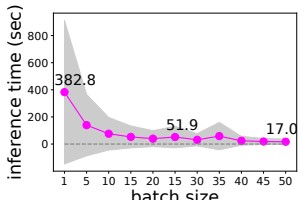

Figure 4: The inference time with different batch sizes.

### 4.2 Reasoning on the 3D-World: Solving CLEVR-Hans problems

**Data.** The CLEVR-Hans data set (Stammer et al., 2021) contains confounded CLEVR (Johnson et al., 2017) images, and each image is associated with a class label. The CLEVR-Hans3 data set has *three* classes, and the CLEVR-Hans7 data set has *seven* classes.

**Model.** We adopted Slot Attention (Locatello et al., 2020a) as a visual perception module and used a set prediction architecture, where each slot representation is fed to MLPs to predict attributes.

**Pre-training.** The slot attention model was pre-trained following (Locatello et al., 2020a) using the set prediction setting on the CLEVR (Johnson et al., 2017) data set. In the concept learning process, we trained `rightside`, `leftside`, and `front` using the scene data in the CLEVR data set. We generated 10k positive and negative examples for each concept, respectively.

**Baselines.** The considered baselines are the ResNet34-based CNN model (Hu et al., 2016), and the Neuro-Symbolic model (NeSy) (Stammer et al., 2021). The NeSy model was trained in *two* different settings: (1) training using classification rules (NeSy-default), and (2) training using classification rules and example-based explanation labels (NeSy-XIL).

**Results.** Table 2 shows the classification accuracy in the CLEVR-Hans data sets. The results of baselines have been presented in (Stammer et al., 2021). In the CLEVR-Hans3 data set, NSFR achieved more than $98\%$ in each split. In the CLEVR-Hans7 data set, NSFR achieved more than $92\%$, that is $>$ NeSy-Default. Note that, NeSy-XIL model exploits example-based labels about the explanation, whereas NeSy-Default and NSFR do not. Thus NeSy-XIL outperforms NSFR marginally. The empirical result shows that NSFR (i) handles different types of the perception models (YOLO and Slot Attention), (ii) can effectively handle 3D images, and more importantly, (iii) is robust to confounded data if the classification rules are available in the form of logic programs.

### 4.3 Fast Inference by Batch Computation

We show that NSFR can perform fast inference by batch computation. Figure 4 shows the inference time with different batch sizes in Kandinsky data sets. We change the batch size from $1$ to $50$ by increments of $5$ and run the experiment in each Kandinsky data set. The magenta line represents mean running time, and the shade represents the standard deviation over the data sets. The empirical result shows that NSFR can perform fast reasoning using batch computation, which is the essential nature of deep neural networks.

## 5 Conclusion and Future Work

We proposed Neuro-Symbolic Forward Reasoner (NSFR), a novel framework for object-centric reasoning tasks. NSFR perceives raw input images using an object-centric model, converts the output into the probabilistic ground atoms, and performs the differentiable forward-chaining inference. Furthermore, NSFR supports batch computation. Thus it combines the perception module and the reasoning module seamlessly. In our experiments, NSFR outperformed conventional CNN-based models in 2D *Kandinsky patterns* and 3D *CLEVR-Hans* data sets, where the classification rules are defined on the high-level concepts. There are several avenues for future work. If we set the clause weights as trainable parameters, NSFR can perform structure learning of logic programs from visual inputs, which is a promising way of extending Inductive Logic Programming and differentiable approaches. Likewise, if we set the parameters of the perception model as trainable parameters, NSFR can train perception models with logical constraints.

ETHICS STATEMENT

With our work, we have shown that we can seamlessly combine symbolic and sub-symbolic systems. Combining neural models with symbolic models can lead to better generalization and handle a wider variety of problems.The major impact that our work aims is enabling coherent quantitative inquiries that encompass multiple data dimension types across object-centric reasoning tasks. This can have several implications on studying how scientific fields evolve and can produce validated signatures predictive of the emergence and success of new fields or discoveries. The results can also be leveraged to create metrics and methods to estimate the innovation potential of scientific enterprises. To the best of our knowledge, our study does not raise any ethical, privacy or conflict of interest concerns.

REPRODUCIBILITY STATEMENT

Upon acceptance, an official GitHub repository will be made public, containing the code of NSFR, and scripts to reproduce the experiments and generate data sets. In addition to this, architectural details and hyper-parameters are included in the appendix. Preliminary code will be uploaded upon submission. Lastly, details on the evaluation metrics and relevant data sets, including the relevant symbolic rules, are given in the main text as well as the appendix.

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

# A    NOTATION

## A.1    NOTATION

| Term | Explanation |
|---|---|
| $\perp$ | a special atom that is always false |
| $\top$ | a special atom that is always true |
| $\mathtt{p}/(n, [\mathtt{dt_1}, \ldots, \mathtt{dt_n}])$ | a neural predicate |
| $\mathtt{p(X, Y)}$ | an atom |
| $\mathtt{p(X, Y) : -q(X, Y)}.$ | a clause |
| $\mathtt{dt}$ | a datatype, e.g., $\mathtt{color}$, $\mathtt{shape}$ |
| $v_{\mathtt{p}}$ | valuation function |
| $\mathcal{T}$ | a set of constants |
| $\mathcal{T}_{\mathtt{dt}}$ | a set of constants of datatype $\mathtt{dt}$, i.e., $dom(\mathtt{dt})$ |
| $\mathcal{O}$ | a set of object constants |
| $\mathcal{A}$ | a set of attribute constants |
| $\mathcal{X}$ | a set of input constants |
| $\mathcal{V}$ | a set of variables |
| $\mathcal{C}$ | a set of clauses |
| $\mathcal{G}$ | a set of ground atoms |
| $\mathcal{P}$ | a set of predicates $\mathcal{P}$ |
| $\mathcal{L}$ | a language $(\mathcal{P}, \mathcal{T}, \mathcal{V})$ |
| $\mathcal{E}_{\mathcal{L}}$ | a set of attribute encoding on language $\mathcal{L}$ |
| $\theta$ | a substitution |
| $\mathbf{x}$ | a vector |
| $\mathbf{X}$ | a tensor |
| $\mathbf{Z} \in \mathbb{R}^{B \times E \times D}$ | an object-centric representation |
| $\mathbf{Z}^{(i)} \in \mathbb{R}^{B \times D}$ | an object-centric representation of the $i$-th object, i.e., $\mathbf{Z}_{:,i,:}$ |
| $\mathbf{A}_{\mathtt{attr}}$ | one-hot encoding of $\mathtt{attr}$ |
| $\mathbf{V}^{(0)} \in \mathbb{R}^{B \times G}$ | an initial valuation vector in batch |
| $\mathbf{V}^{(t)} \in \mathbb{R}^{B \times G}$ | a valuation vector in batch at time-step $t$ |
| $\mathbf{I} \in \mathbb{R}^{C \times G \times S \times L}$ | an index tensor |
| $\mathbf{\Phi}$ | the parameter in object-centric models |
| $\mathbf{\Theta}$ | the parameter in neural predicates |
| $\mathbf{W} \in \mathbb{R}^{M \times C}$ | clause weights |
| $G$ | the number of ground atoms, i.e., $|\mathcal{G}|$ |
| $C$ | the number of clauses, i.e., $|\mathcal{C}|$ |
| $B$ | batch size of the input |
| $S$ | the maximum number of substitutions for existentially quantified variables |
| $L$ | the maximum number of body atoms |
| $E$ | the maximum number of objects in an image |
| $D$ | the dimension object-centric representation |
| $M$ | the size of logic program, i.e., the number of distinct clause weights |
| $N$ | input dimension |
| $\gamma$ | the smooth parameter for softor function |
| $softor_d^{\gamma}$ | the softor fucntion for dimension $d$ with a smooth parameter $\gamma$ |
| $stack_d$ | the stack method to concatenate tensors along a new dimension $d$ |
| $\odot$ | element-wise multiplication between two tensors |

Table 3: Notations in this paper.

## B    Facts Converting Algorithm

---

**Algorithm 1** Convert the object-centric representation into probabilistic facts

---

**Input:** object-centric representation $\mathbf{Z} \in \mathbb{R}^{B \times E \times D}$, a set of ground atoms $\mathcal{G}$, background knowl-
edge $\mathcal{B}$, a set of one-hot encoding of attributes $\mathcal{E}_{\mathcal{L}}$

 1: initialize $\mathbf{V}^{(0)} \in \mathbb{R}^{B \times |\mathcal{G}|}$ as a zero tensor
 2: **for** $F_i = \mathrm{p}(\mathtt{t_1}, \ldots, \mathtt{t_n}) \in \mathcal{G}$ **do**
 3:     **if** p is a neural predicate **then**
 4:         **for** $\mathtt{t_j}$ in $[\mathtt{t_1}, \ldots, \mathtt{t}_n]$ **do**
 5:             $\mathbf{T}_j = f_{to\_tensor}(\mathtt{t}_j; \mathbf{Z}, \mathcal{E}_{\mathcal{L}})$
 6:         $\mathbf{V}^{(0)}_{:,i} = v_{\mathrm{p}}(\mathbf{T}_1, \ldots, \mathbf{T}_n)$ // call the valuation function for neural predicate p
 7:     **else if** $F_i \in \mathcal{B}$ **then**
 8:         $\mathbf{V}^{(0)}_{:,i} = 1.0$ // set background knowlege
 9: **return** $\mathbf{V}^{(0)}$

---

## C    Data sets in Experiments

We make the data sets used in the paper available at: `https://bit.ly/3FhTeOY`.

### C.1    Kandinsky Dataset Summary

|  | TwoPairs | ThreePairs | Closeby | Red-Triangle | Online-Pair | 9-Circles |
|---|---|---|---|---|---|---|
| Combination | Yes (Hard) | Yes (Hard) | No | Yes (Easy) | Yes (Easy) | Yes (Hard) |
| Spatial | No | No | Yes (Easy) | Yes (Easy) | Yes (Hard) | No |
| objects | 4 | 6 | 4 | 6 | 5 | 9 |
| training data | 5000 | 5000 | 5000 | 5000 | 5000 | 1000 |
| test/val data | 2000 | 2000 | 2000 | 2000 | 2000 | 500 |

Table 4: Kandinsky patterns data set summary. Each data set is different in terms of combinatorial or spatial patterns.

### C.2    Classification Rules

**TwoPairs**    The pattern for positive figures is: *"the Kandinsky Figure has two pairs of objects with the same shape. In one pair, the objects have the same colors in the other pair different colors. Two pairs are always disjunct, i.e., they do not share objects."*. This can be represented as a logic program containing *four* clauses:

```
1 kp(X):-in(O1,X),in(O2,X),in(O3,X),in(O4,X),same_shape_pair(O1,O2),
      same_color_pair(O1,O2),same_shape_pair(O3,O4),diff_color_pair(O3,O4).
2 same_shape_pair(X,Y):-shape(X,Z),shape(Y,Z).
3 same_color_pair(X,Y):-color(X,Z),color(Y,Z).
4 diff_color_pair(X,Y):-color(X,Z),color(Y,W),diff_color(Z,W).
```

**ThreePairs**    The pattern for positive figures is: *"the Kandinsky Figure has three pairs of objects with the same shape. In one pair, the objects have the same colors in other pairs different colors. Three pairs are always disjunct, i.e., they do not share objects."*. This can be represented as a logic program containing *four* clauses:

```
1 kp(X):-in(O1,X),in(O2,X),in(O3,X),in(O4,X),in(O5,X),in(O6,X),
      same_shape_pair(O1,O2),same_color_pair(O1,O2),same_shape_pair(O3,O4),
      diff_color_pair(O3,O4),same_shape_pair(O5,O6),diff_color_pair(O5,O6).
2 same_shape_pair(X,Y):-shape(X,Z),shape(Y,Z).
3 same_color_pair(X,Y):-color(X,Z),color(Y,Z).
4 diff_color_pair(X,Y):-color(X,Z),color(Y,W),diff_color(Z,W).
```

**Closeby**   The pattern for positive figures is: *"the Kandinsky Figure has a pair of objects that are close by each other."*. This can be represented as a logic program containing *one* clause:

```
1 kp(X) :- in(O1,X),in(O2,X),closeby(O1,O2).
```

**Red-Triangle**   The pattern for positive figures is: *"the Kandinsky figure has a pair of objects that are close by each other, and the one object of the pair is a red triangle, and the other object has a different color and different shape."*. This can be represented as a logic program containing *three* clauses:

```
1 kp(X) :- in(O1,X),in(O2,X),closeby(O1,O2),color(O1,red),shape(O1,triangle
    ),diff_shape_pair(O1,O2),diff_color_pair(O1,O2).
2 diff_shape_pair(X,Y) :- shape(X,Z),shape(Y,W),diff_shape(Z,W).
3 diff_color_pair(X,Y):-color(X,Z),color(Y,W),diff_color(Z,W).
```

**Online/Pair**   The pattern for positive figures is: *"the Kandinsky figure has five objects that are aligning on a line, and it contains at least one pair of objects that have the same shape and the same color."*. This can be represented as a logic program containing *three* clauses:

```
1 kp(X) :- in(O1,X),in(O2,X),in(O3,X),in(O4,X),in(O5,X),online(O1,O2,O3,O4,
    O5),same_shape_pair(O1,O2),same_color_pair(O1,O2).
2 same_shape_pair(X,Y) :- shape(X,Z),shape(Y,Z).
3 same_color_pair(X,Y) :- color(X,Z),color(Y,Z).
```

**9-Circles**   The pattern for positive figures is: *"the Kandinsky figure has three red objects, three blue objects, and three yellow objects."*[1]. This can be represented as a logic program containing *four* clauses:

```
1 kp(X):-has_red_triple(X),has_yellow_triple(X),has_blue_triple(X).
2 has_red_triple(X):-in(O1,X),in(O2,X),in(O3,X),color(O1,red),color(O2,red)
    ,color(O3,red).
3 has_yellow_triple(X):-in(O1,X),in(O2,X),in(O3,X),color(O1,yellow),color(
    O2,yellow),color(O3,yellow).
4 has_blue_triple(X):-in(O1,X),in(O2,X),in(O3,X),color(O1,blue),color(O2,
    blue),color(O3,blue).
```

**CLEVR-Hans3**   The data set has *three* classification rules. We refer to (Stammer et al., 2021) for more details. We used the following logic program in NSFR:

```
1 kp1(X):-in(O1,X),in(O2,X),size(O1,large),shape(O1,cube),size(O2,large),
    shape(O2,cylinder).
2 kp2(X):-in(O1,X),in(O2,X),size(O1,small),material(O1,metal),shape(O1,cube
    ),size(O2,small),shape(O2,sphere).
3 kp3(X):-in(O1,X),in(O2,X),size(O1,large),color(O1,blue),shape(O1,sphere),
    size(O2,small),color(O2,yellow),shape(O2,sphere).
```

**CLEVR-Hans7**   The data set has *seven* classification rules. We refer to (Stammer et al., 2021) for more details. We used the following logic program in NSFR:

```
1 kp1(X):-in(O1,X),in(O2,X),size(O1,large),shape(O1,cube),size(O2,large),
    shape(O2,cylinder).
2 kp2(X):-in(O1,X),in(O2,X),size(O1,small),material(O1,metal),shape(O1,cube
    ),size(O2,small),shape(O2,sphere).
3 kp3(X):-in(O1,X),in(O2,X),in(O3,X),color(O1,cyan),front(O1,O2),front(O1,
    O3),color(O2,red),color(O3,red).
4 kp4(X):-in(O1,X),in(O2,X),in(O3,X),in(O4,X),size(O1,small),color(O1,green
    ),size(O2,small),color(O2,brown),size(O3,small),color(O3,purple),size
    (O4,small).
5 kp5(X):-has_3_spheres_left(X).
```

---

[1]The 9-circles data set has been public as a challenge data set (https://github.com/human-centered-ai-lab/dat-kandinsky-patterns). The original data set contains counterfactual examples that falsify a simple hypothesis. We excluded the counterfactual examples to simplify the problem.

```
6  kp5(X):-has_3_spheres_left(X),has_3_metal_cylinders_right(X).
7  kp6(X):-has_3_metal_cylinders_right(X).
8  kp7(X):-in(O1,X),in(O2,X),size(O1,large),color(O1,blue),shape(O1,sphere),
       size(O2,small),color(O2,yellow),shape(O2,sphere).
9  has_3_spheres_left(X):-in(O1,X),in(O2,X),in(O3,X),shape(O1,sphere),shape(
       O2,sphere),shape(O3,sphere),leftside(O1),leftside(O2),leftside(O3).
10 has_3_metal_cylinders_right(X):-in(O1,X),in(O2,X),in(O3,X),shape(O1,
       cylinder),shape(O2,cylinder),shape(O3,cylinder),material(O1,metal),
       material(O2,metal),material(O3,metal),rightside(O1),rightside(O2),
       rightside(O3).
```

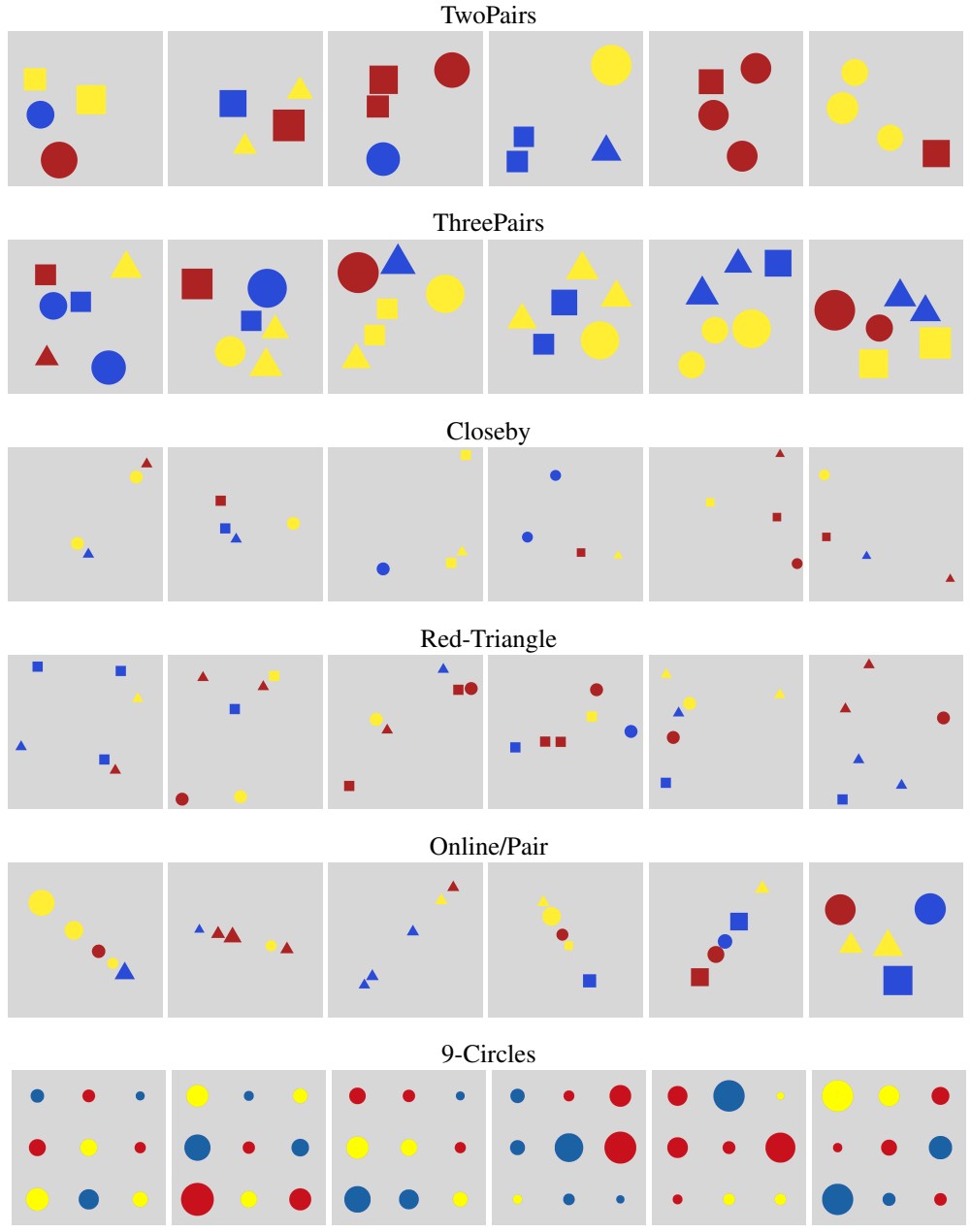

Figure 5: Training examples in each Kandinsky data set. The left three images are positive examples, and the right three images are negative examples.

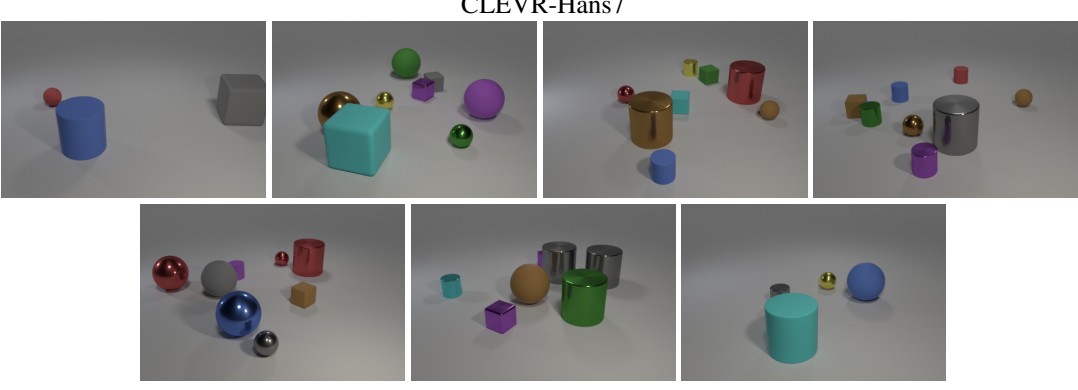

Figure 6: Examples in the CLEVR-Hans data set. CLEVR-Hans3 has *three* classes and CLEVR-Hans7 has *seven* classes, respectively. Each example represents each class in the data set.

# D   OBJECT-CENTRIC PERCEPTION MODELS IN EXPERIMENTS

We used different object-centric perception models for Kandinsky and CLEVR-Hans data sets. In this section, we describe model details and the pre-training setting. All experiments were performed on one NVIDIA A100-SXM4-40GB GPU with 40 GB of RAM.

## D.1   YOLO FOR KANDINSKY DATA SET

We used YOLOv5[2] model, whose implementation is publicly available. We adopted the YOLOv5s model, which has 7.3M parameters.

**Pre-training**   We generated $15,000$ pattern-free figures for training, $5000$ figures for validation. Figure 7 shows the statistics of the pre-training data set. The class labels and positions are generated randomly. The original image size is $620 \times 620$, and resized into $128 \times 128$. The label consists of the class labels and the bounding box for each object. The class label is generated by the combination of the shape and the color of the object, e.g., *red circle* and *blue square*. The number of classes is $9$. Each image contains at least 2 objects, and at most 10 objects. Figure 8 shows the confusion matrix for the pre-trained model. The confusion matrix of the pre-training data set of the YOLOv5 model. The pre-trained YOLOv5 model classifies the objects correctly in Kandinsky patterns.

## D.2   SLOT ATTENTION FOR CLEVR-HANS DATA SET

We used the same setup as (Stammer et al., 2021). In the preprocessing, we downscaled the CLEVR-Hans images to visual dimensions $128 \times 128$ and normalized the images to lie between $-1$ and $1$. For training the slot-attention module, an object is represented as a vector of binary values for the shape, size, color, and material attributes and continuous values between $0$ and $1$ for the $x$, $y$, and $z$ positions. We refer to (Locatello et al., 2020a) for more details.

---

[2]https://github.com/ultralytics/yolov5

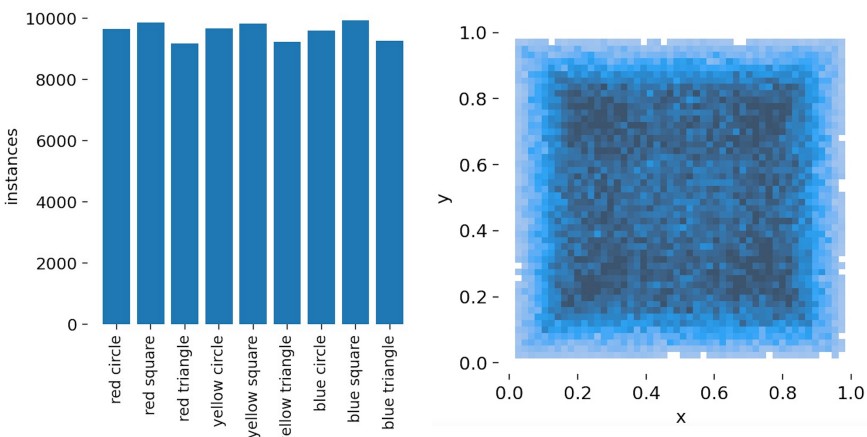

Figure 7: The statistics of the pre-training data set in Kandinsky Patterns tasks. The distribution of the class label (left) and the distribution of the position of the objects (right). The class labels and the positions are generated randomly.

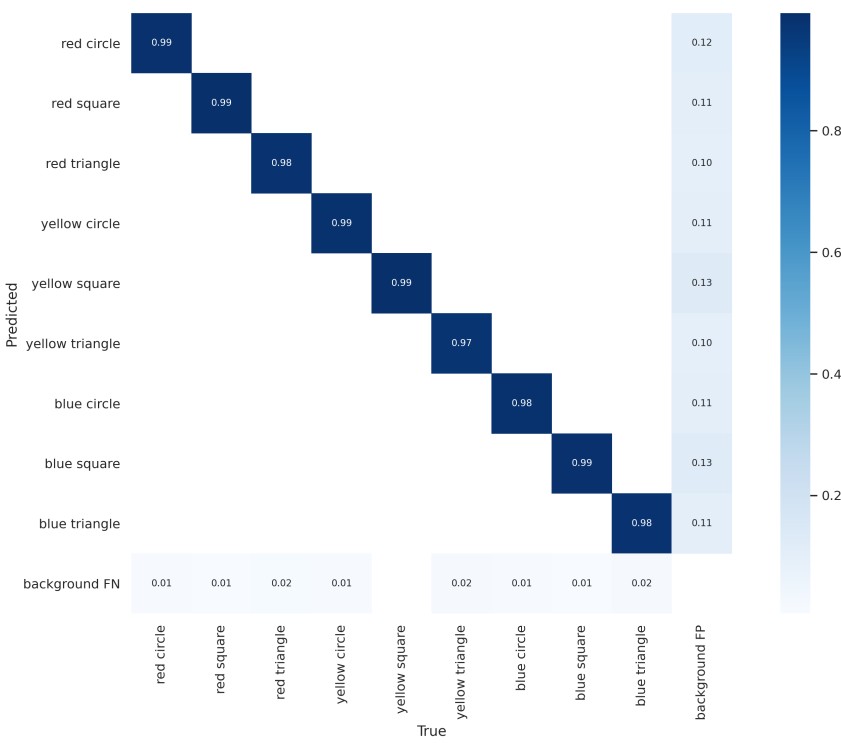

Figure 8: The confusion matrix of the YOLOv5 model in the test split after the pre-training. The trained YOLOv5 model classifies the objects in Kandinsky figures correctly.

# E LANGUAGES IN EXPERIMENTS

In this section, we show the language settings we used in each data set.

## E.1 DATA TYPES AND CONSTANTS

Tab. 5 and Tab. 6 shows the constants and their data types for each Kandinsky and CLEVR data set, respectively.

| Datatype | Terms |
|---|---|
| image | img |
| object | obj1, obj2, ..., obj9 |
| color | red, blue, yellow |
| shape | square, circle, triangle |

Table 5: Datatype and constants in Kandinsky data set.

| Datatype | Terms |
|---|---|
| object | obj1, obj2, ..., obj9 |
| color | cyan, blue, yellow, purple, red, green, gray, brown |
| shape | sphere, cube, cylinder |
| size | large, small |
| material | rubber, metal |
| side | right, left |

Table 6: Datatype and constants in CLEVR data set.

## E.2 PREDICATES

| Predicate | Explanation |
|---|---|
| kp/[image] | The image belongs to a pattern. |
| same_shape_pair/[object, object] | The two objects have a same shape. |
| same_color_pair/[object, object] | The two objects have a same color. |
| diff_shape_pair/[object, object] | The two objects have different shapes. |
| diff_color_pair/[object, object] | The two objects have different colors. |

Table 7: Predicates in Kandinsky data set.

| Neural Predicate | Explanation |
|---|---|
| in/[object, image] | The object is in the image. |
| shape/[object, shape] | The object has the shape of shape. |
| color/[object, color] | The object has the color of color. |
| closeby/[object, object] | The two objects are located close by each other. |
| online/[object, ..., object] | The objects are aligned on a line. |

Table 8: Neural predicates in Kandinsky data set.

| Predicate | Explanation |
|---|---|
| kpi/[image] | The image belongs to the $i$-th pattern. |
| same_shape_pair/[object, object] | The two objects have a same shape. |
| same_color_pair/[object, object] | The two objects have a same color. |
| has$_3$_spheres_left/[iamge] | The image has three spheres on the left side. |
| has_3_metal_cylinders_right/[iamge] | The image has three metal cylinders on the right side. |

Table 9: Predicates in CLEVR-Hans data set.

| Neural Predicate | Explanation |
|---|---|
| in/[object, image] | The object is in the image. |
| shape/[object, shape] | The object has the shape of shape. |
| color/[object, color] | The object has the color of color. |
| material/[object, material] | The object has the color of material. |
| size/[object, size] | The object has the color of size. |
| leftside/[object] | The object is on leftside. |
| rightside/[object] | The object is on rightside. |
| front/[object, object] | The first object is front of the second object. |

Table 10: Neural predicates in CLEVR-Hans data set.

### E.3 BACKGROUND KNOWLEDGE

In *TwoPairs, ThreePairs*, and *Red-Triangle* data sets, we prepared background knowledge for NSFR about predicate diff_color and diff_shape as $\mathcal{B}$ = {diff_color(red, blue), diff_color(blue, red), diff_color(red, yellow), diff_color(yellow, red), diff_color(blue, yellow), diff_color(yellow, blue), diff_shape(circle, square), diff_shape(square, circle), diff_shape(circle, triangle), diff_shape(triangle, circle), diff_shape(square, triangle), diff_shape(triangle, square)}.

## F VALUATION FUNCTIONS

### F.1 VALUATION FUNCTIONS FOR KANDINSKY PATTERNS WITH YOLO

In our experiments, the output format of the YOLO model is as in the following table.

| index | 0 | 1 | 2 | 3 | 4 | 5 | 6 | 7 | 8 | 9 | 10 |
|---|---|---|---|---|---|---|---|---|---|---|---|
| attribute | $x_1$ | $y_1$ | $x_2$ | $y_2$ | red | yellow | blue | square | circle | triangle | *objectness* |

Here, $(x_1, y_1)$ and $(x_1, y_2)$ is the coordinates of the top-left and bottom-right points of the bounding box. Each attribute dimension contains each probability.

The valuation function for each neural predicate is shown in Tab. 11. Tensor $\mathbf{Z}^{(i)}_{center}$ for predicate closeby represents the center coordinate of the bounding box for the $i$-th object. Function $f_{linear}$ for predicate online computes the closed-form solutions of linear regression in batch and returns the error values.

| Atom | Valuation Function |
|---|---|
| in(obj1, img) | $v_{\texttt{in}}(\mathbf{Z}^{(1)}, \mathbf{X}) = \mathbf{Z}^{(1)}_{:,10} \in \mathbb{R}^B$ // return the objectness |
| shape(obj1, circle) | $v_{\texttt{shape}}(\mathbf{Z}^{(1)}, \mathbf{A}_{\texttt{circle}}) = sum_1(\mathbf{Z}_{1,8:11} \odot \mathbf{A}_{\texttt{circle}}) \in \mathbb{R}^B$ |
| color(obj2, red) | $v_{\texttt{color}}(\mathbf{Z}^{(2)}, \mathbf{A}_{\texttt{red}}) = sum_1(\mathbf{Z}_{:,4:7} \odot \mathbf{A}_{\texttt{red}}) \in \mathbb{R}^B$ |
| closeby(obj1, obj2) | $v_{\texttt{closeby}}(\mathbf{Z}^{(1)}, \mathbf{Z}^{(2)}) = \sigma(norm_0(\mathbf{Z}^{(1)}_{center} - \mathbf{Z}^{(2)}_{center}); \mathbf{w}) \in \mathbb{R}^B$ |
| online(obj1, ..., obj5) | $v_{\texttt{online}}(\mathbf{Z}^{(1)}, ..., \mathbf{Z}^{(5)}) = \sigma\left(f_{linear}(\mathbf{Z}^{(1)}, ..., \mathbf{Z}^{(5)}); \mathbf{w}\right) \in \mathbb{R}^B$ |

Table 11: Valuation functions for each neural predicate in Kandinsky data set. Each neural predicate is associated with a valuation function. In the forward-chaining reasoning step, the probability for each ground atom is computed using the valuation function. The parameterized neural predicates are trained using the concept examples.

### F.2 VALUATION FUNCTIONS FOR CLEVR-HANS WITH SLOT ATTENTION

In our experiments, the output format of the slot attention model is as in the following table.

| index | 0 | 1 | 2 | 3 | 4 | 5 | 6 | 7 | 8 | 9 | 10 |
|-------|---|---|---|---|---|---|---|---|---|---|----|
| attribute | *objectness* | $x$ | $y$ | $z$ | sphere | cube | cylinder | large | small | rubber | metal |

| 11 | 12 | 13 | 14 | 15 | 16 | 17 | 18 |
|----|----|----|----|----|----|----|----|
| cyan | blue | yellow | purple | red | green | gray | brown |

The valuation function for each neural predicate is shown in Tab.12.

| Atom | Valuation Function |
|------|-------------------|
| `in(obj1,img)` | $v_{\texttt{in}}(\mathbf{Z}^{(1)}, \mathbf{X}) = \mathbf{Z}^{(1)}_{:,0} \in \mathbb{R}^B$ // return objectness |
| `shape(obj1,sphere)` | $v_{\texttt{shape}}(\mathbf{Z}^{(1)}, \mathbf{A}_{\texttt{sphere}}) = sum_1(\mathbf{Z}^{(1)}_{:,4:7} \odot \mathbf{A}_{\texttt{circle}}) \in \mathbb{R}^B$ |
| `size(obj1,large)` | $v_{\texttt{size}}(\mathbf{Z}^{(1)}, \mathbf{A}_{\texttt{large}}) = sum_1(\mathbf{Z}^{(1)}_{:,7:9} \odot \mathbf{A}_{\texttt{large}}) \in \mathbb{R}^B$ |
| `material(obj1,metal)` | $v_{\texttt{material}}(\mathbf{Z}^{(1)}, \mathbf{A}_{\texttt{metal}}) = sum_1(\mathbf{Z}^{(1)}_{:,9:11} \odot \mathbf{A}_{\texttt{circle}}) \in \mathbb{R}^B$ |
| `color(obj1,red)` | $v_{\texttt{color}}(\mathbf{Z}^{(1)}, \mathbf{A}_{\texttt{red}}) = sum_1(\mathbf{Z}^{(1)}_{:,11:19} \odot \mathbf{A}_{\texttt{red}}) \in \mathbb{R}^B$ |
| `leftside(obj1)` | $v_{\texttt{leftside}}(\mathbf{Z}^{(1)}) = \sigma(\mathbf{Z}^{(1)}_{:,1}; \mathbf{w}) \odot \mathbf{Z}^{(1)}_{:,0} \in \mathbb{R}^B$ // the objectness is multiplied |
| `rightside(obj1)` | $v_{\texttt{rightside}}(\mathbf{Z}^{(1)}) = \sigma(\mathbf{Z}^{(1)}_{:,1}; \mathbf{w}) \odot \mathbf{Z}^{(1)}_{:,0} \in \mathbb{R}^B$ // the objectness is multiplied |
| `front(obj1,obj2)` | $v_{\texttt{front}}(\mathbf{Z}^{(1)}, \mathbf{Z}^{(2)}) = \sigma\left(\left[\mathbf{Z}^{(1)}_{:,1:4}, \mathbf{Z}^{(2)}_{1:4}\right]; \mathbf{w}\right) \odot \mathbf{Z}^{(1)}_{:,0} \odot \mathbf{Z}^{(2)}_{:,0} \in \mathbb{R}^B$ |

Table 12: Valuation functions for each neural predicate in CLEVR-Hans data set. Each neural predicate is associated with a valuation function. In the forward-chaining reasoning step, the probability for each ground atom is computed using the valuation function. The parameterized neural predicates are trained using the concept examples.

## G    DETAILS OF TENSOR ENCODING

**Preliminaries.** A *unifier* for the set of expressions $\{A_1, \ldots, A_n\}$ is a substitution $\theta$ such that $A_1\theta = A_2\theta = \ldots = A_n\theta$, written as $\theta = \sigma(\{A_1, \ldots, A_n\})$, where $\sigma$ is a *unification function*. A unification function returns the (most general) unifier for the expressions if they are unifiable. Decision function $\bar{\sigma}(\{A_1, \ldots, A_n\})$ returns a Boolean value whether or not $A_1, \ldots, A_n$ are unifiable.

**Dealing with Existentially Quantified Variables.** We extend the differentiable forward-chaining inference to deal with a flexible number of existentially quantified variables. For example, let clause $C_i = \texttt{kp(X)} : -\texttt{in(O1,X)}.$. The clause has the existentially quantified variable $\texttt{O1}$. First we consider the possible substitutions for $\texttt{O1}$, e.g., $\{\texttt{O1/obj1}, \texttt{O1/obj2}\}$. For ground atom $\texttt{kp(img)}$, by applying these substitutions to the body atoms, we get the grounded clauses as:

$$\texttt{kp(img)} : -\texttt{in(obj1,img)}. \tag{6}$$
$$\texttt{kp(img)} : -\texttt{in(obj2,img)}. \tag{7}$$

In this case, the maximum number of substitutions $S = 2$. Using these grounded clauses, we can build the index tensor for the differentiable inference function.

Formally, for each pair of $F_j \in \mathcal{G}$ and $C_i = A : -B_1 \ldots B_n \in \mathcal{C}$, let $\theta_{head} = f_{unify}(\{F_j, A\})$. For body atoms $\{B_1, \ldots, B_n\}$, we compute $\{B_1^*, \ldots, B_n^*\}$ where $B_i^* = B_i\theta_{head}$. Let $\mathcal{V}_{i,j} = V(B_1^*)\cup, \ldots, \cup V(B_n^*)$, where $V$ is a function that returns a set of variables in the input atom. The set of possible substitutions $\mathcal{S}$ is computed as:

$$\mathcal{S}_{i,j} = \{\texttt{X/t} \mid \texttt{X} \in \mathcal{V}_{i,j} \wedge \texttt{t} \in dom(\texttt{dt})\}, \tag{8}$$

where $\texttt{dt}$ is a datatype for constant $\texttt{t}$, which can be determined from the definition of the predicate. The maximum number of substitutions for body atoms (subgoals) equals to the maximum size of the set $S = \max_{i,j} |\mathcal{S}_{i,j}|$. If there is a lot of possible substitutions for body atoms, NSFR can consume a lot of memories because the size of the index tensor is proportional to $S$. In the computation of possible substitutions, we assume that different constants are substituted for different variables, respectively.

**Tensor Encoding (Formal).** We build a tensor that holds the relationships between clauses $\mathcal{C}$ and ground atoms $\mathcal{G}$. We assume that $\mathcal{C}$ and $\mathcal{G}$ are an ordered set, i.e., where every element has its own index. Let $L$ be the maximum body length in $\mathcal{C}$, $C = |\mathcal{C}|$, and $G = |\mathcal{G}|$. Index tensor $\mathbf{I} \in \mathbb{N}^{C \times G \times S \times L}$ contains the indices of the ground atoms to compute forward inferences. Intuitively, $\mathbf{I}_{i,j,k,l}$ is the index of the $l$-th ground atom in the $i$-th clause to derive the $j$-th ground atom with the $k$-th substitution for existentially quantified variables.

For clause $C_i = A : -B_1, ..., B_n \in \mathcal{C}$ and ground atom $F_j \in \mathcal{G}$, let the body substitutions $\mathcal{S}_{i,j}$. We compute tensor $\mathbf{I} \in \mathbb{R}^{C \times G \times S \times L}$:

$$\mathbf{I}_{i,j,k,l} = \begin{cases} I_{\mathcal{G}}(B_l \theta_k) \text{ if } \bar{\sigma}(\{A, F_j\}) \wedge l \leq n \\ I_{\mathcal{G}}(\top) \text{ if } \bar{\sigma}(\{A, F_j\}) \wedge k > n \\ I_{\mathcal{G}}(\bot) \text{ if } \neg\bar{\sigma}(\{A, F_j\}) \end{cases} , \qquad (9)$$

where $\theta_k \in \mathcal{S}_{i,j}$, $0 \leq l \leq L - 1$, $\theta = \sigma(\{A, F_j\})$, and $I_{\mathcal{G}}(F)$ returns the index of $F$ in $\mathcal{G}$. If clause head $A$ and ground atom $F_j$ are unifiable, then we put the index of subgoal $B_k\theta$ into the tensor (line 1 in Eq. 9). If the clause has fewer body atoms than the longest clause in $\mathcal{C}$, we fill the gap with the index of $\top$ (line 2 in Eq. 9). If clause head $A$ and ground atom $G_j$ are not unifiable, then we place the index of $\bot$ (line 3 in Eq. 9).

# H    LEARNING AND REASONING ON NSFR

In NSFR, we adopt the curriculum learning approach as illustrated in Figure 9.

**Step1: Training visual-perception model** The visual-perception module is trained on pattern-free figures. Each figure is generated randomly without any patterns. Depending on the specific type of the visual-perception model, each object has its annotation.

**Step2: Concept Learning** Neural predicates are trained using the pre-trained perception module. Then the parameterized neural predicates are trained on figures prepared for each concept.

**Step3: Reasoning on figures on patterns** On the reasoning step, NSFR performs reasoning using the trained visual-perception model and neural predicates. The logical rules are given as weighted clauses.

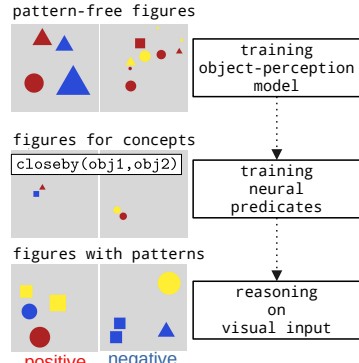

Figure 9: Curriculum learning and reasoning in NSFR.

# I    DETAILS ON THE $softor_d^\gamma$ FUNCTION

### I.0.1    THE $softor_d^\gamma$ FUNCTION

In the differentiable inference process, NSFR often computes logical *or* for probabilistic values. Taking *max* repeatedly can violate the gradients flow. The $softor_d^\gamma$ function approximates the *or* computation softly. The key idea is to use the log-sum-exp technique. We define the $softor_d^\gamma$ function as follows:

$$softor_d^\gamma(\mathbf{X}) = \frac{1}{S}\gamma \log\left(sum_d \exp\left(\mathbf{X}/\gamma\right)\right), \qquad (10)$$

where $sum_d$ is the sum function for tensors along dimension $d$, and

$$S = \begin{cases} 1.0 & \text{if } max\left(\gamma \log sum_d \exp\left(\mathbf{X}/\gamma\right)\right) \leq 1.0 \\ max\left(\gamma \log sum_d \exp\left(\mathbf{X}/\gamma\right)\right) & \text{otherwise} \end{cases} \qquad (11)$$

The normalization term ensures that the $softor_d^\gamma$ function returns a normalized probabilistic values. The dimension $d$ specifies the dimension to ta be removed.

A popular choice is the *probabilistic sum* function: $f_{prob\_sum}(\mathbf{X}, \mathbf{Y}) = \mathbf{X} + \mathbf{Y} - \mathbf{X} \odot \mathbf{Y}$, which was adopted in (Evans & Grefenstette, 2018) and (Jiang & Luo, 2019). We compare these functions with the proposed approach. Fig.11 visualizes each function, and Fig.11 visualizes the difference of each function from the original *logical or* function, i.e., the max function. With a sufficiently small smooth parameter, the $softor_d^\gamma$ function approximates the original max function. In our experiments in Kandinsky and CLEVR-Hans data sets, we consistently set $\gamma = 0.01$.

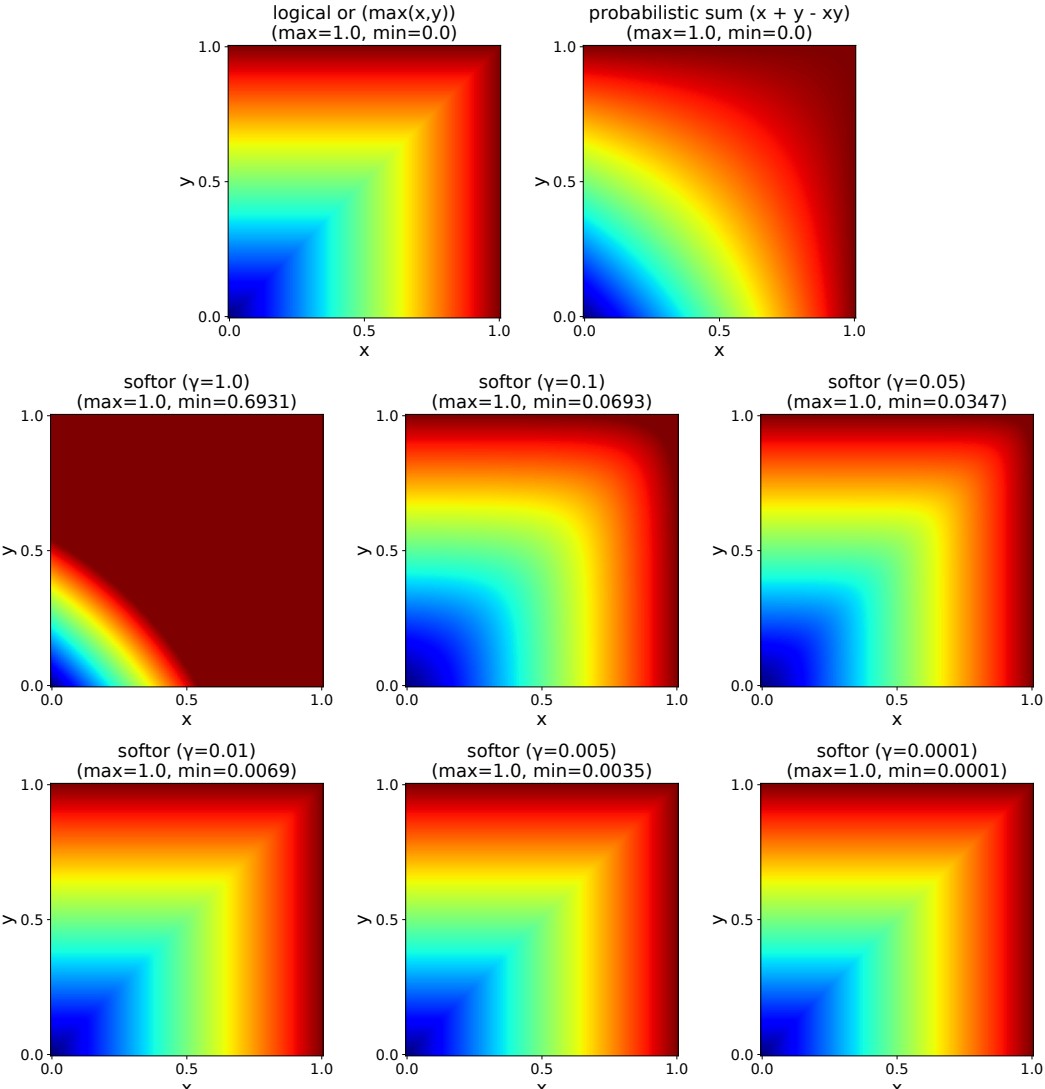

Figure 10: The visualization of various *or* functions. The maximum and minimum values for each image is shown on top. The $softor_d^\gamma$ function with a sufficiently small smooth parameter approximates the logical *or* function for probabilistic values.

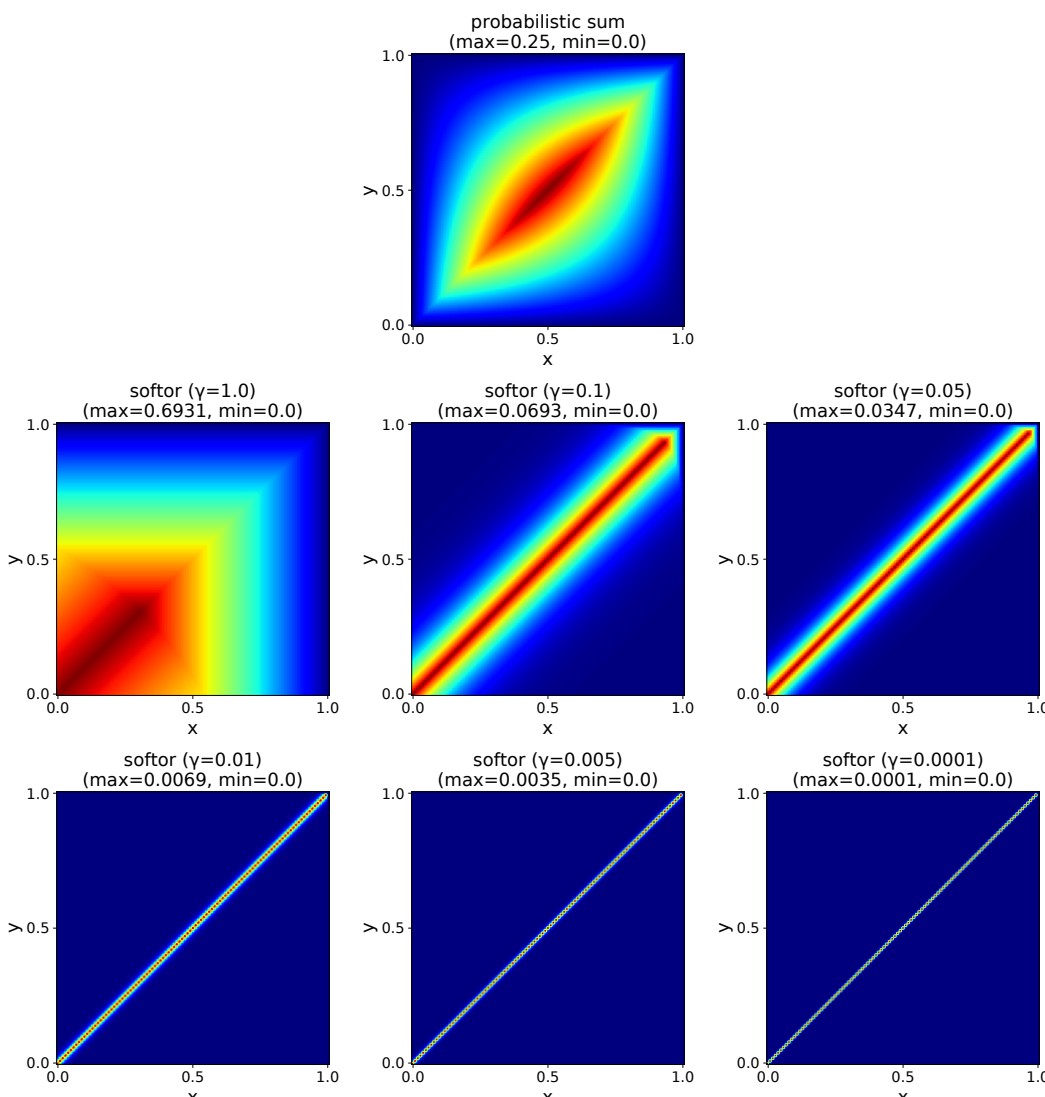

Figure 11: The visualization of the difference between the original *or* function and other *or* functions, i.e., probabilistic sum and $softor_d^\gamma$. The maximum and minimum values in each image is shown on top. The $softor_d^\gamma$ function with a sufficiently small smooth parameter approximates the logical *or* function for probabilistic values.

