# OpenReview forum: "Neuro-Symbolic Forward Reasoning"
_ICLR.cc/2022/Conference — ICLR 2022 Submitted_

### Official Review · Reviewer_Jwkf · 2021-10-23

**Correctness:** 2
**Technical Novelty And Significance:** 2
**Empirical Novelty And Significance:** 2
**Recommendation:** 3
**Confidence:** 4

**Main Review:**

The paper tackles an important problem, using an approach with great promise (neurosymbolic approach). However, I do have serious misgivings about the paper.

If the authors had achieved the same performance without so many hand-designed features and hand-coded rules - in short, with fewer built-in priors - it would be much more impressive. But the proposed system appears to carry out very little learning (apart from the pre-training of the encoder), and incorporates a large number of hand-designed elements tailored for the benchmarks. Most glaringly, as far as I can tell, the forward-chaining reasoning module imports a hand-written set of clauses that essentially encode the solution to the problem at hand. For example, in Section C.2, the authors give the clauses used by their system for the CLEVR-Hans3 dataset (Stammer, et al (2021)), which include:

kp1(X):-in(O1,X),in(O2,X),size(O1,large),shape(O1,cube),size(O2,large),shape(O2,cylinder).

Not only does this directly encode class 1 of the dataset, it also, by construction (ie: by “cheating”), solves the confounding issue that the CLEVR-Hans dataset was designed to bring out. In this class, the confounding factor in the training set is the colour of the large cube. In the training set, it happens to be always grey. But in the test set, it is sometimes another colour. There is no in-principle way to work out the “correct” rule from the training set alone, which is why Stammer et al incorporate user feedback to revise their model after training. By contrast, the present authors’ rule omits the confounding factor (colour) from the outset. So it’s little wonder they do well on the evaluation. Regrettably, this seems to me like a fundamental flaw to the paper - although, of course, I might be missing something, and I’m keen to hear the authors’ response here.

Relatedly, on p.4, the authors write “We make the minimum assumption that the perception function takes an image and returns a set of object-centric vectors, where each of the vectors represents each object. For simplicity, we assume that each dimension of the vector represents the probability of the attributes for each object.” And in Section F.2, the authors give a table showing the “output format of the slot attention model”. This format perfectly represents the features needed for the CLEVR-Hans benchmark. I believe the output from the slot attention module is obtained in this format, following Locatello et al, thanks to a supervised learning stage that depends on the availability of ground truth labels corresponding to shape, size, colour, etc. (Is this correct?) It seems to me that this builds in half the solution to the problem from the outset. (I grant that Stammer et al make the same move, but their paper solves a harder problem; see above.) In effect, it means that the low-level object attributes (eg: “sphere”), which correspond to atoms in the clauses (eg: shape(obj1,sphere)), are hand-designed rather than learned.

It might be helpful to see a (qualitative, at least) comparison with other architectures that tackle similar problems (relational reasoning about images), using architectures that are also to some degree neurosymbolic. Many of these have far fewer built-in priors than the present architecture, such as Santoro et al (2017) (not so neurosymbolic), Asai (2019), or Shanahan et al (2020). The latter two architectures have a number of similarities with the present one, but learn objects, features, and relations end-to-end from scratch. The architecture of Stammer et al (2021) also learns the reasoning part from scratch (using a set transformer), without domain-specific knowledge of the relevant rules.

Asai, M. Unsupervised grounding of plannable first-order logic representation from images. In International Conference on Automated Planning and Scheduling, 2019.

Shanahan, M., Nikiforou, K., Creswell, A., Kaplanis, C., Barrett, D. and Garnelo, M. An explicitly relational neural network architecture. In International Conference on Machine Learning, pp. 8593-8603, 2020.

Santoro, A., Raposo, D., Barrett, D. G., Malinowski, M.,Pascanu, R., Battaglia, P., and Lillicrap, T.   A simple neural network module for relational reasoning.  In Advances in Neural Information Processing Systems, pp. 4974–4983, 2017.

I was initially a bit puzzled why the authors repeatedly refer to the batch dimension, both in their formal presentation and in characterising their contribution, since it’s customary to omit the batch dimension for simplicity. However, a sentence at the end of section 3.4.3 states that the “value function computes probablility in batch”. In other words, the batch matters, and can’t be simply omitted for clarity. Is this correct? As I was also wondering what the justification was for interpreting value functions probabilistically, maybe the authors should make this point a bit more prominently, and give a bit more explanation. Indeed, I’m still confused by the role of probability in the architecture.

It seems that the architecture is not end-to-end differentiable, as the algorithm for converting object-centric representations to probabilistic facts appears not to be differentiable. This isn't necessarily a problem, although it precludes fine-tuning the pre-trained encoder for a given task. But maybe the authors could confirm my understanding here.

**Summary Of The Paper:**

The paper tackles the problem of how to incorporate symbolic reasoning into a differentiable, deep learning architecture. The authors present an architecture comprising a pre-trained, slot-based encoder, and a differentiable clausal reasoner. They apply it to visual, relational reasoning problems, evaluating the architecture on two datasets: Kandinsky patterns and CLEVR-Hans, achieving promising performance figures.

**Summary Of The Review:**

This is an important line of research, in my view, so the work is of potential value. However, I have major concerns about the extent to which the architecture relies on hand-designed features and rules. As well as limiting the value of the work, in its present state, I believe this issue also invalidates some of the evaluation (on CLEVR-Hans). But I am open to having my mind changed by the rebuttal.

---

> ### Author Response · Authors · 2021-11-15
> **Response to Reviewer Jwkf (Part 1/2)**
>
> We thank the reviewer for the comments. We answer the raised questions pointwise.
>
> *Q1: If the authors had achieved the same performance without so many hand-designed features and hand-coded rules - in short, with fewer built-in priors - it would be much more impressive. But the proposed system appears to carry out very little learning.*
>
> A1: I agree with the fact that NSFR carries out little learning compared to the other neuro-symbolic frameworks. We would clarify here that our focus is on the reasoning part, not on the (structure) learning part. NSFR can be extended to a more learning-based framework because the differentiable forward reasoning [1,2], which NSFR adopts to perform reasoning, has been developed for Inductive Logic Programming [3,4], which carries out the structure learning of logic programs from examples. Therefore, NSFR is the first step towards building a framework that performs learning and reasoning, combining neural networks with logical reasoning and induction techniques.
> Moreover, we would argue here that Probabilistic Logic Programming [5] has been developed in many works focusing on reasoning and inference architecture because learning algorithms are developed based on the reasoning mechanism. Therefore, we believe that also in the case of neuro-symbolic learning, the reasoning architecture should be established first.
>
>
> *Q2: By contrast, the present authors’ rule omits the confounding factor (colour) from the outset. So it’s little wonder they do well on the evaluation. Regrettably, this seems to me like a fundamental flaw to the paper.*
>
> A2: Our evaluation setting can be considered as follows. If we have prior knowledge as a set of rules, we can make use of the rules to reason. Therefore, NSFR and NeSy-XIL [6] very well extend each other. If training data of confounding behavior is available, then NeSy-XIL can be applied, else if the users have already provided some rules to reason, NSFR can be applied directly.
>
>
> *Q3: As I was also wondering what the justification was for interpreting value functions probabilistically, maybe the authors should make this point a bit more prominently, and give a bit more explanation. Indeed, I’m still confused by the role of probability in architecture.*
>
> A3: We have mainly two reasons to use probabilities in NSFR.
> First, the differentiable forward-chaining is defined as on probabilistic ground atoms [1,2]. Therefore, to leverage the differentiable forward-chaining reasoning module, the valuation function outputs probabilistic values.
> Second, the probabilistic interpretation of logic programs has rich literature and a theoretical foundation [7]. The approach has been established with a lot of applications in Statistical Relational Learning [8]. We can utilize arguments and findings from many works in the field by giving the probabilistic interpretation in the architecture.
>
>
> *Q4: I believe the output from the slot attention module is obtained in this format, following Locatello et al, thanks to a supervised learning stage that depends on the availability of ground truth labels corresponding to shape, size, colour, etc. (Is this correct?) It seems to me that this builds in half the solution to the problem from the outset.*
>
> A4: Yes, it is correct. The slot attention module has been trained using the ground truth labels.
> From the learning perspective, NSFR requires a lot of supervision. However, it is still not trivial to perform reasoning using given knowledge in a differentiable way for visual inputs that contain several objects. As we argued in A1, we believe that the foundation of the reasoning mechanism on visual inputs based on the differentiable logic is the first step to building a solid framework to perform learning using both neural-based and symbolic inductive learning techniques.
>
>
> *Q5: It might be helpful to see a (qualitative, at least) comparison with other architectures that tackle similar problems (relational reasoning about images), using architectures that are also to some degree neurosymbolic.*
>
> A5: Thank you for the suggestion and the list of papers. We will consider discussing these papers and performing some comparisons with other neuro-symbolic architectures in the future.

---

> > ### Author Response · Authors · 2021-11-15
> > **Response to Reviewer Jwkf (Part 2/2)**
> >
> > *Q6: the batch matters, and can’t be simply omitted for clarity. Is this correct?*
> >
> > A6: Yes, the batch matters in NSFR. In the logic-based neuro-symbolic frameworks, the parallel computation of the batch of inputs is not trivial. For example, DeepProblog [9] constructs a diagram, which is called Sentential Decision Diagram, for every single query. Therefore, the system processes sequentially each example in a given batch of inputs. NeurASP [10] calls the CPU-based implementation of the Answer Set Programming solver for every single query. [10] reported that NeurASP took 4min 24sec, DeepProbLog took 38min 1sec, and a CNN took 14 sec for 30,000 iterations of inference. This result highlights the fact that the logic-based neuro-symbolic frameworks have a limitation in terms of inference speed. Therefore, speeding up the inference by the tensor-based parallel batch computation is an important contribution.
> > Moreover, previous works on the differentiable forward reasoning [1,2] did not discuss how to perform parallelized batch computation. They assumed that input is a vector. In evidence, the implementation of [1] processes each example in a batch sequentially. NSFR can perform forward reasoning for a batch of visual input in parallel. This feature is important if we combine the reasoning module with neural networks. As shown in the experiment, in Fig. 4, the inference is pretty slow if we process every single image without parallelized batch computation. A promising future application of NSFR would be training neural networks with logical constraints.
> >
> >
> > *Q7: It seems that the architecture is not end-to-end differentiable, as the algorithm for converting object-centric representations to probabilistic facts appears not to be differentiable.*
> >
> > A7: The converting algorithm is differentiable with respect to the parameters in the object-perception model. In the converting process, for each ground atom, NSFR calls a valuation function to compute its probability given input image. To call the valuation functions, we map the constant in first-order logic to a corresponding tensor representation, which is not parameterized. The tensor representations of constants are stored as static tensors and referred to repeatedly from the fact-converter module. Through the converting process, the gradient flow is not violated. We would also note here that, as we argued in A6, NSFR processes the given batch in parallel also in the converting process, which is a vital feature of NSFR compared to previous logic-based Neuro-Symbolic frameworks [9,10]. The facts converter needs to perform the for-loop in terms of the ground atoms, but not in terms of the input images in the batch.
> >
> > Thanks again for the fruitful comments.
> >
> >
> > [1] Hikaru Shindo, Masaaki Nishino, Akihiro Yamamoto: Differentiable Inductive Logic Programming for Structured Examples, AAAI, 2021.
> >
> > [2] Richard Evans, Edward Grefenstette: Learning Explanatory Rules from Noisy Data, JAIR, 2018.
> >
> > [3] Stephen Muggleton.: Inductive logic programming. New Generation Computing 8(4): 295–318, 1991.
> >
> > [4] Andrew Cropper, Sebastijan Dumančić, Stephen H. Muggleton: Turning 30: New Ideas in Inductive Logic Programming, IJCAI, 2020.
> >
> > [5] Daan Fierens, Guy Van den Broeck, Joris Renkens, Dimitar Shterionov, Bernd Gutmann, Ingo Thon, Gerda Janssens, and Luc De Raed: Inference and learning in probabilistic logic programs using weighted Boolean formulas, Theory and Practice of Logic Programming, 2015.
> >
> > [6] Wolfgang Stammer, Patrick Schramowski, Kristian Kersting: Right for the Right Concept: Revising Neuro-Symbolic Concepts by Interacting with their Explanations, CVPR, 2021.
> >
> > [7] Taisuke Sato: A Statistical Learning Method for Logic Programs with Distribution Semantics, ICLP, 1995.
> >
> > [8] Luc De Raedt, Kristian Kersting, Sriraam Natarajan, David Poole: Statistical Relational Artificial Intelligence: Logic, Probability, and Computation. Synthesis Lectures on Artificial Intelligence and Machine Learning. Morgan & Claypool Publishers, 2016.
> >
> > [9] Robin Manhaeve, Sebastijan Dumančić, Angelika Kimmig, Thomas Demeester, Luc De Raedt: DeepProbLog: Neural Probabilistic Logic Programming, NeurIPS, 2018.
> >
> > [10] Zhun Yang, Adam Ishay, Joohyung Lee: NeurASP: Embracing Neural Networks into Answer Set Programming, IJCAI, 2020.

---

> > > ### Comment · Reviewer_Jwkf · 2021-11-22
> > > **Response to response**
> > >
> > > Many thanks to the authors for their answers and clarifications.
> > >
> > > My Q2 was:
> > >
> > > Q2: By contrast, the present authors’ rule omits the confounding factor (colour) from the outset. So it’s little wonder they do well on the evaluation. Regrettably, this seems to me like a fundamental flaw to the paper.
> > >
> > > and the authors' response was:
> > >
> > > A2: Our evaluation setting can be considered as follows. If we have prior knowledge as a set of rules, we can make use of the rules to reason. Therefore, NSFR and NeSy-XIL [6] very well extend each other. If training data of confounding behavior is available, then NeSy-XIL can be applied, else if the users have already provided some rules to reason, NSFR can be applied directly.
> > >
> > > Unfortunately this only serves to confirm my initial understanding, and I'm sorry to say that I do consider this to be a serious issue. It renders the direct compariosn with NeSy and NeSy-XIL in table 2 invalid.
> > >
> > > See also my comment to the response to all reviewers above.

---

> > > > ### Author Response · Authors · 2021-11-28
> > > > **Response to the reviewer**
> > > >
> > > > We would like to point it out to the reviewer that the problems considered in our work i.e. Kandinsky patterns and CLEVR-Hans are so hard that we do not know how basic basic deep neural networks can solve them. With our method i.e. using differential forward chaining we can easily code the solution and thus we have a chance to solve these tricky tasks.
> > > >
> > > > Regarding color as confounder, we would like to point it out to the reviewer that the problem of synthesizing pixels with symbols i.e. constraints is an open problem. Expressing this constraint via logic is what we are trying to achieve with this work.

---

### Official Review · Reviewer_ZLoi · 2021-10-24

**Correctness:** 4
**Technical Novelty And Significance:** 2
**Empirical Novelty And Significance:** 1
**Recommendation:** 3
**Confidence:** 3

**Main Review:**

Pros:
1. A whole system pipeline is developed for neural-symbolic reasoning. If the authors can open source their code, this will be a good contribution to the community.
2. The system's advantage (mainly in term of accuracy) is demonstrated in 2 selected datasets.

Cons:
1. I can see how this work is different from a few other works on neural-symbolic visual reasoning, but I cannot see how this work innovate on previous ideas. The authors say that previous work (e.g. Amizadeh et al., 2020) either does not have a differentiable forward reasoner, or an object-centric module. But I think works such as Amizadeh et al., 2020 and neural-symbolic concept learner have both of these modules? I would appreciate if the authors can state more clearly the difference between their work and the previous works.
2. I feel the evaluation is not enough to gain more insights into the strength of the work. First of all, only two synthetic datasets are used for evaluation. And for the CLEVR-dataset, the authors only reported on CLEVR-Hans dataset, which has very few baselines to compare against. Results on the standard CLEVR datasets will allow us to gain more insights into the model's capability. Experiments on real-world datasets (such as GQA) will also boost confidence in the work. The evaluation should also go beyond simple classification accuracies. One strength of neuro-symbolic model is its capability to generalize beyond the training distributions. I would encourage the authors to at least performs some experiments on compositional generalization.

**Summary Of The Paper:**

The paper proposes a differential neuro-symbolic visual reasoning framework that combines an object representation extractor and a forward reasoner.

**Summary Of The Review:**

A technically solid paper that lacks a bit of novelty and falls shorts in evalaution.

---

> ### Author Response · Authors · 2021-11-15
> **Response to Reviewer ZLoi (Part 1/2)**
>
> We thank the reviewer for the comments. We answer the raised questions pointwise.
>
> *Q1: How this work innovate on previous ideas?*
>
> A1: Briefly, NSFR has advantages with respect to two perspectives. (i) NSFR is a logic-based neuro-symbolic framework, which is not dedicated to VQA tasks, and (ii) NSFR establishes a seamless combination with object-centric neural networks in terms of parallelized batch computation.
> - (i)  NSFR is a neuro-symbolic framework based on first-order logic. The advantage of adopting first-order logic is mainly two-fold. First, logic has rich literature as a language to describe human knowledge on computers. As demonstrated in the experiments, the knowledge can be given in a form of rules. NSFR can reason using the output of the object-centric perception model in a differentiable manner. We argue about the difference from the VQA-based neuro-symbolic models in the answer to Q2.
> - (ii) Previous logic-based neuro-symbolic frameworks, such as DeepProblog [1] and NeurASP [2] have a limitation in terms of the speed of inference. The NeurASP paper reported that DeepProblog and NeurASP take way longer compared to convolutional neural networks to perform inference. Moreover, these architectures do not support object-centric perception models.  NSFR supports object-centric perception models and also supports fast parallelized inference for a given batch of visual input.
>
> *Q2: But I think works such as Amizadeh et al., 2020 and neural-symbolic concept learner have both of these modules? I would appreciate if the authors can state more clearly the difference between their work and the previous works.*
>
> A2:
> - VQA-based Neuro-Symbolic frameworks, e.g., Amizadeh [3] and NS-CL [4], are dedicated to VQA tasks. In the frameworks, the structure of symbolic programs is determined by the linguistic structure of the questions. The symbolic programs are generated based on the questions which are given as natural language sentences. In Kandinsky Patterns and CLEVR-Hans tasks, we do not assume that we have questions in the form of natural language sentences. Therefore, VQA-based neuro-symbolic frameworks cannot be applied directly.
> - In essence, NSFR is an interface between high-level human cognition and visual inputs. NSFR employs first-order logic as its symbolic program. The machine-learning paradigm based on first-order logic, which is called Inductive Logic Programming (ILP) [5,6], is a well-established approach. It has different features compared to neural-based approaches, e.g., it can learn a generalized solution from small data sets [7]. Differentiable forward reasoning has been developed for ILP [7,8]. Thus building a pipeline between neural-based models to percept objects and differentiable forward reasoning is the first step to build a consistent framework to perform inductive learning from visual images leveraging the symbolic learning technique. We note that probabilistic logic programming [9] has been developed based on many works focusing on the reasoning part because the learning algorithms are developed on the reasoning mechanism. Therefore, we believe that also in the case of neuro-symbolic learning, the reasoning architecture should be established first.
>
>
> *Q3: Experiments on real-world datasets (such as GQA) will also boost confidence in the work. The evaluation should also go beyond simple classification accuracies.*
>
> A3: Thank you for the suggestions. Again VQA is not the focus of our paper but we will consider these data sets when we extend our method to the question-answering domain.

---

> > ### Author Response · Authors · 2021-11-15
> > **Response to Reviewer ZLoi (Part 2/2)**
> >
> > *Q4: I would encourage the authors to at least perform some experiments on compositional generalization.*
> >
> > A4: Thank you for the suggestion. We would like to address in future the compositional generalization using e.g. the CURI data set [10]. We would note here, in this work, our aim is to establish the reasoning mechanism for object-centric reasoning tasks using neural networks and differentiable logic. As we argued in A2, we believe that NSFR can be extended to more learning-based architecture.
> >
> >
> > [1] Robin Manhaeve, Sebastijan Dumančić, Angelika Kimmig, Thomas Demeester, Luc De Raedt: DeepProbLog: Neural Probabilistic Logic Programming, NeurIPS, 2018.
> >
> > [2] Zhun Yang, Adam Ishay, Joohyung Lee: NeurASP: Embracing Neural Networks into Answer Set Programming, IJCAI, 2020.
> >
> > [3]  Saeed Amizadeh, Hamid Palangi, Oleksandr Polozov, Yichen Huang, Kazuhito Koishida: Neuro-Symbolic Visual Reasoning: Disentangling "Visual" from "Reasoning", ICML, 2020.
> >
> > [4] Jiayuan Mao, Chuang Gan, Pushmeet Kohli, Joshua B. Tenenbaum, Jiajun Wu: The Neuro-Symbolic Concept Learner: Interpreting Scenes, Words, and Sentences From Natural Supervision, ICLR, 2019.
> >
> > [5] Stephen Muggleton.: Inductive logic programming. New Generation Computing 8(4): 295–318, 1991.
> >
> > [6] Andrew Cropper, Sebastijan Dumančić, Stephen H. Muggleton: Turning 30: New Ideas in Inductive Logic Programming, IJCAI, 2020.
> >
> > [7] Richard Evans, Edward Grefenstette: Learning Explanatory Rules from Noisy Data, JAIR, 2018.
> >
> > [8] Hikaru Shindo, Masaaki Nishino, Akihiro Yamamoto: Differentiable Inductive Logic Programming for Structured Examples, AAAI, 2021.
> >
> > [9] Daan Fierens, Guy Van den Broeck, Joris Renkens, Dimitar Shterionov, Bernd Gutmann, Ingo Thon, Gerda Janssens, and Luc De Raed: Inference and learning in probabilistic logic programs using weighted Boolean formulas, Theory and Practice of Logic Programming, 2015.
> >
> > [10] Ramakrishna Vedantam, Arthur Szlam, Maximilian Nickel, Ari Morcos, Brenden Lake: CURI: A Benchmark for Productive Concept Learning Under Uncertainty, ICML, 2021.

---

### Official Review · Reviewer_iyKK · 2021-10-28

**Correctness:** 4
**Technical Novelty And Significance:** 1
**Empirical Novelty And Significance:** 1
**Recommendation:** 5
**Confidence:** 4

**Main Review:**

Strengths

Using differentiable reasoning methods towards this task makes much more sense than deep learning models. Such a method should be ideally more interpretable, verifiable, robust, and generalizable. Neuro-symbolic reasoning has received notable attention recently and I think this paper is another step towards general AI with good reasoning capability.

The formality in this work should be encouraged. Compared to earlier works in this topic, the paper clearly defines the notions and related concepts.

The work is well presented with important notions explained with timely and concise examples. I like the flow the work in general: it's easy to understand with exemplified motivation and illustrative graphics.

Weaknesses

My major concern with this work is the novelty. Specifically, I'm not sure what the contribution of this work is with respective to earlier papers. Differentiable reasoning with / without object-centric representation has been investigated in works that could be dated back to DeepProbLog [1], NS-VQA [2], NS-CL [3], NLM [4]. [1] and [4] discuss how logic reasoning can be combined with differentiable learning (batched or not) and [2] and [3] use object-based representation to solve visual reasoning problems with clearly defined programs. Therefore, I'm confused on the specific novelty of this work. It reads like another work using differentiable reasoning without much new contribution to the problem.

A common issue with logic-based reasoning is the prior knowledge required. Before learning, the designer needs to supply the algorithm with space of atoms, the background knowledge (rules), etc. How does it compare with other deep learning models that can be trained without them?

While I do not doubt the effectiveness of the method, the improvement over previous methods looks limited. The object-based YOLO+MLP model looks bad in complex scenes but could it be because of the amount of data you use? I assume the space for complex scenes is much larger for simpler scenes and with the fixed amount of training data, complex scenes will be less well-covered than simpler ones. And on simpler scenes, YOLO+MLP is not really bad. On the CLEVR-Hans experiments, NSFR is not always the best either.

[1] DeepProbLog: Neural Probabilistic Logic Programming
[2] Neural-Symbolic VQA: Disentangling Reasoning from Vision and Language Understanding
[3] The Neuro-Symbolic Concept Learner: Interpreting Scenes, Words, and Sentences From Natural Supervision
[4] Neural Logic Machines


**Summary Of The Paper:**

The paper proposes a method for differentiable reasoning using soft first-order logic. The key idea is to combine forward reasoning with object-based deep learning. In particular, after the perception process from a pre-trained object detector, the proposed Neuro-Symbolic Forward Reasoner (NSFR) converts the object-based representation into probabilistic atoms and performs forward reasoning using differentiable logic. In Kandinsky and CLEVR-Hands datasets, the method shows slightly improved performance over neural methods and other symbolic approaches.

**Summary Of The Review:**

I appreciate the merits in this work as mentioned above. However, I will give a temporary weak reject due to its limited novelty.

I have read the reviews and the authors' response. The authors acknowledge some general issues and point out  differences with earlier works, though I do not think is significant enough. Syncing with other reviewers, I decide to keep my initial rating.

---

> ### Author Response · Authors · 2021-11-15
> **Response to Reviewer iyKK (Part 1/2)**
>
> We thank the reviewer for the comments. We answer the raised questions pointwise.
>
>
> *Q1: what is the contribution of this work is with respect to earlier papers. Differentiable reasoning with / without object-centric representation has been investigated in works that could be dated back to DeepProbLog [1], NS-VQA [2], NS-CL [3], NLM [4]. Briefly, [1] and [4] discuss how logical reasoning can be combined with differentiable learning (batched or not), and [2] and [3] use object-based representation to solve visual reasoning problems with clearly defined programs.*
>
> A1:
>  We describe the differences and advantages of NSFR with respect to the mentioned previous work.
> - DeepProblog does not support neural networks that return object-centric representation. Therefore it is not trivial to apply the method to object-centric reasoning tasks such as Kandinsky patterns and CLEVR-Hans. More importantly, the inference part is not fully differentiable in DeepProblog in terms of batch computation. DeepProblog builds a diagram, which is called Sentential Decision Diagram, for each inference query. That means the inference part cannot be implemented in a fully GPU-friendly way. In general, neural networks can process a batch of input in parallel, but DeepProblog needs to process each input sequentially. NSFR can be parallelized for a given batch of inputs. This feature is crucial to combine logical reasoning with neural networks and perform inference on large-scale data sets.
> - NS-VQA and NS-CL are dedicated to VQA tasks. In architectures for VQA tasks, the symbolic programs are determined by the linguistic structure of the questions. The symbolic programs are interconnected with given questions. In Kandinsky patterns and CLEVR-Hans tasks, we do not assume that we have questions in the form of natural language sentences. NSFR assumes that the symbolic program is consistent.
> Moreover, NSFR employs first-order logic as its symbolic program. There are multiple benefits to employing first-order logic. First, the programs can be learned by the symbolic learning technique, which is called Inductive Logic Programming (ILP) [5,6]. ILP has some advantages compared to neural networks, e.g., it can learn a generalized solution from small data. Differentiable forward reasoning has been developed to integrate ILP with neural networks [7,8]. Thus building a pipeline between neural-based models to perceive objects and differentiable forward reasoning is the first step to build a consistent framework to perform inductive learning from visual images leveraging both neural networks and ILP techniques.
> - In essence, NLM employs a different type of reasoning in their architecture. The differentiable forward-chaining reasoning approach [7,8], which includes NSFR, is a more symbolic logic-based approach. Therefore, the symbolic inductive learning techniques can be incorporated naturally.  Moreover, NLM does not address the visual inputs that contain several objects.
>
>
> *Q2: Before learning, the designer needs to supply the algorithm with space of atoms, the background knowledge (rules), etc. How does it compare with other deep learning models that can be trained without them?*
>
> A2: This is an open question for all of the neuro-symbolic learning frameworks using logic as their language.
> In general, users need to give a language bias, which determines the search space in a logic-based framework [9].
> This is the main weakness in logic-based learning approaches when users do not have much information or knowledge about the task. The positive aspect is that users can inject their knowledge into the model when some prior knowledge is available. NSFR provides a way to feed some prior knowledge in object-centric reasoning tasks.

---

> > ### Author Response · Authors · 2021-11-15
> > **Response to Reviewer iyKK (Part 2/2)**
> >
> > *Q3: The improvement over previous methods looks limited. The object-based YOLO+MLP model looks bad in complex scenes but could it be because of the amount of data you use? I assume the space for complex scenes is much larger.*
> >
> > A3: Thank you for pointing it out. We will address the problem with a larger data set to assess the performance of the neural-based benchmarks. We would note here that one of the major motivations of the logic-based neuro-symbolic approach is to establish a data-efficient learning method [8]. We believe that NSFR could be a contribution towards building such a data-efficient inductive learning framework from visual input combining neural networks with symbolic logic.
> >
> >
> > [1] Robin Manhaeve, Sebastijan Dumančić, Angelika Kimmig, Thomas Demeester, Luc De Raedt: DeepProbLog: Neural Probabilistic Logic Programming, NeurIPS, 2018.
> >
> > [2] Kexin Yi, Jiajun Wu, Chuang Gan, Pushmeet Kohli, Antonio Torralba, Joshua B. Tenenbaum: Neural-Symbolic VQA: Disentangling Reasoning from Vision and Language Understanding, NeurIPS, 2018.
> >
> > [3] Jiayuan Mao, Chuang Gan, Pushmeet Kohli, Joshua B. Tenenbaum,Jiajun Wu: The Neuro-Symbolic Concept Learner: Interpreting Scenes, Words, and Sentences From Natural Supervision, ICLR, 2019.
> >
> > [4] Honghua Dong, Jiayuan Mao, Tian Lin, Chong Wang, Lihong Li, Denny Zhou: Neural Logic Machines, ICLR, 2019.
> > 295–318, 1991.
> >
> > [5] Stephen Muggleton.: Inductive logic programming. New Generation Computing 8(4): 295–318, 1991.
> >
> > [6] Andrew Cropper, Sebastijan Dumančić, Stephen H. Muggleton: Turning 30: New Ideas in Inductive Logic Programming, IJCAI, 2020.
> >
> > [7] Hikaru Shindo, Masaaki Nishino, Akihiro Yamamoto: Differentiable Inductive Logic Programming for Structured Examples, AAAI, 2021.
> >
> > [8] Richard Evans, Edward Grefenstette: Learning Explanatory Rules from Noisy Data, JAIR, 2018.
> >
> > [9] Hilde Ade, Luc DeRaedt, Maurice Bruynooghe:  Declarative Bias for Specific-to-General ILP Systems, Machine Learning, 20, 119-154, 1995.

---

### Official Review · Reviewer_hJBE · 2021-11-02

**Correctness:** 2
**Technical Novelty And Significance:** 3
**Empirical Novelty And Significance:** Not applicable
**Recommendation:** 3
**Confidence:** 4

**Main Review:**

The caption on Figure 1 is not clear.

From Page 1 it is not clear what a Kandinsky pattern is.

“it is difficult, if not impossible, to solve object-centric reasoning tasks such as Kandinsky patterns due to several underlying challenges: (i) the perception of the objects from the raw inputs and (ii) the reasoning on the attributes and the relations to capture the complex patterns”. It’s not clear what the object-centric reasoning task in question is here? I would also argue that a Neural-Symbolic VQA (Kexin Yi et al.) type model could solve this type of problem? Since it has been applied very successfully to Clevr. What makes Kandisky patterns more challenging? This is not made clear in the paper.

Page 2: “It computes the set of ground atoms” <— It is not clear what **it** refers to here.

It would be helpful in the introduction to talk about the level of supervision in the model, just at a high level.

Related work:
“They either do not employ a differentiable forward reasoner or miss objet-centric learning in the end-to-end reasoning architecture.” It would be helpful to be clear about which methods fall in to which category. It is also unclear how this work build on previous work.

Further,  “Deep Compositional Question Answering with Neural Module Networks”,  and “NeuroSymbolic Concept Learner” use differentiable components. It would be good to include some comparison here between differentiable components and a differentiable end-to-end reasoning architecture.

“NSFR encapsulates different object-perception models, thus allows us to choose a proper model depending on the situation and the problem to be solved.” Does this mean that NSFR can learn both with and without supervision? However, later in the paper is appears that you do need supervision? Please clarify.

From the introduction it is not clear exactly what the contributions of the paper are and what the related work is? The related work section is very hard to follow.

Object-centric Perception: It is not clear what kinds of supervision are needed for this process? In the facts converter the authors assume that “The output of the visual-perception module is already factorized in terms of objects.” How is this assumption enforced?

In figure 3 and Example 2 it is not clear what a “valuation function” is. What is $v_{color}$ defined only for $A_{red}$ and $v_{shape}$ only for $A_{circle}$.

Facts Converter: The Facts Converter is not clearly described, however, Figure 3 was helpful for understanding the process.

Where do the ground atoms in Figure 3 come from? It appears that you can have significantly more ground atoms than are necessary to describe the two scene? Do you need to define new ground atom for each task? It would be helpful to discuss this.

Experiments:

It would be helpful to be very explicit about the supervision provided for each of the baselines compared to the NSFR model?

Table 1: Why not compare to other Neuro-Symbolic methods such as NeuroSymbolic concept learner or Neural-Symbolic VQA? Or even using a relation net (A simple neural network module for relational reasoning, Santoro et al)?

Table 1: The ResNet is trained on how many samples? 5k or 15k or 20k? Perhaps a smaller model would perform better? From the appendix it looks like the ResNet is trained on 5k examples, while the NSFR model gets an **extra** 15k examples to train the visual perception module on, so this does not look like a fair comparison. Further, NeuroSymbolic VQA is able to learn complex visual reasoning from only 7k examples. A relation net can learn from 10k sort-of-clevr images.

Table 1 and 2: How many runs were done? What is the variance? It would be helpful to see the training curves to better understand the overfitting in table 1.

Existing Neuro-Symbolic papers look at more than classification accuracy. “Neural-Symbolic VQA”, “NeuroSymbolic Concept learner,” “Deep Compositional Question Answering with Neural Module Networks”, “Learning by Abstraction: The Neural State Machine” and “An Explicitly Relational Neural Network Architecture” all show interesting transfer capabilities like generalising to unseen colour combinations or to problems with more objects than those seen during training.

What are the losses used for training? These are not described anywhere?

Typos:

Page 1: Fig. (a) and Fig. (b) missing figure number.

Page 2: “Facts converter converts” —> **The* facts converter

Page 2: “Finally, differentiable reasoning” —> Finally, **the** differentiable

Page 3: “The typical approach is the object detection (or supervised) approach such as FasterRCNN”. It is not clear what its meant here.

Page 3: “where the models acquire the ability of object-perception without or fewer annotations”. This is not clear.

**Summary Of The Paper:**

The authors propose a model for doing forwards reasoning. There are three parts: The model decomposes a scene into object, converts that to a grounded symbolic representation and then performs differentiable reasoning. The authors show classification performance on two datasets.

While the authors propose a Neuro-Symbolic  model, they do not compare to any of the state of the art neuro-symbolic models such as NeuralSymbolic VQA or NeuroSymbolic concept learner for one of the datasets and only consider one neuro-symbolic model for the other task.

A number of implementation details are also not clear. It’s unclear how much supervision is needed and what form that supervision takes?

It is also not clear what the advantages of the proposed model are over existing models? This is not discussed clearly in the paper and there are no significant experimental results.

The high level direction of this work is very interesting and indeed Neuro-Symbolic models do offer some advantages over deep learning models. However, the authors have failed to discuss or demonstrate any of these advantages (more details below).

**Summary Of The Review:**

My main concerns with this paper are (1) there are not sufficient details to understand how the model works in terms of the supervision needed and the losses computed, (2) it is not clear what the motivations and contributions are (the authors only claim to show improvement in overall accuracy on very simple tasks), (3) the experimental results are not sufficient. The tasks appear to be quite simple (even though ResNets perform badly this appears to be due to lack of data, but the paper does not discuss specifically learning in low data regimes); the authors could have compared to stronger baselines; some of the comparisons may be unfair (see above) and the authors only show classification accuracy.

---

> ### Author Response · Authors · 2021-11-15
> **Response to Reviewer hJBE (Part 1/3)**
>
> We thank the reviewer for the comments. We answer the raised questions pointwise.
>
> *Q1: How much supervision do we need?*
>
> A1: In the current form, NSFR requires the following supervisions:  (i) the supervision to train object-perception model, (ii) the supervision to train neural predicates (concept learning), and (iii) rules to reason. The details for each supervision are as follows.
> - (i) Depending on the architecture of the object-perception model, NSFR requires different types of labels to train. For example, to train the YOLO [1] model, NSFR requires the class label and the bounding box for each object. To train the slot attention model using the set prediction architecture [2], NSFR requires the class label for each object. We note that, in this training phase, the perception model can be trained on the pattern-free data set. In our experiments, we used randomly-generated pattern-free figures for Kandinsky patterns, and we trained the slot attention module using the original CLEVR data set without any classification patterns.
> - (ii) In NSFR, some parameterized neural predicates are trained from examples. For example, the “closeby” relation in the Kandinsky data set and the “front” relation in the CLEVR dataset. NSFR needs supervision for each concept to be trained. The concept examples can take different forms, e.g., in the Kandinsky data set, we used Kandinsky figures, and in the CLEVR data set we used the scene data.
> - (iii) In this work, we assumed that we provide the set of rules to reason.
>
>
> *Q2: What form that supervision takes?*
>
> A2: The supervision is performed in a curriculum learning format (for more details see appendix H, Page 22). We performed the training steps as described in A1.
>
>
> *Q3: What are the advantages of the proposed model over existing models?*
>
> A3: We would like to argue mainly from two aspects, (i) advantages over VQA-based Neuro-Symbolic models and (ii) advantages over logic-based Neuro-Symbolic models.
> - (i) Compared to VQA-based Neuro-Symbolic models, e.g. NS-CL [3], NSFR has the advantage of using first-order logic as its main language. In VQA-based models, the symbolic programs are determined by the natural language sentences but NSFR does not have the assumption. Therefore, NSFR can be naturally combined with symbolic learning techniques, which is called Inductive Logic Programming (ILP) [4,5]. We note that the differentiable forward-chaining reasoning has been developed for ILP [6,7]. We believe that NSFR is the first step to establishing the framework to combine the ILP with object-centric visual input. We would note here that our aim is not solving VQA tasks but establishing a logic-based neuro-symbolic framework that can perform reasoning on visual inputs and be incorporated with symbolic inductive learning techniques in the future.
> - (ii) Compared to previous logic-based Neuro-Symbolic models, e.g., DeepProblog [8] and NeurASP [9], NSFR has the advantage of the seamless combination of the neural networks with the reasoning module in terms of parallelized batch computation. For example, DeepProblog builds a diagram for each query input to perform reasoning, and NeurASP calls a CPU-based implementation of the Answer Set Programming solver. These processes are not GPU-friendly and thus batch processing is an important property of our method. This limitation in the previous work would be a problem to train neural networks on large data sets using logical constraints. Every module in NSFR is implemented as an instance of the neural network class in PyTorch (torch.nn). As shown in Fig. 4, NSFR can perform reasoning rapidly using batch computation. NSFR computed logical entailments for 5k visual inputs in 17 seconds on average on the Kandinsky patterns data set.
>
> *Q4: What the object-centric reasoning task in question is here?*
>
> A4: Here, the object-centric task means the object-centric reasoning problem, which is defined in Def. 1 in section 3.1, Pg. 3.
>
>
> *Q5: I would also argue that a Neural-Symbolic VQA (Kexin Yi et al.) type model could solve this type of problem?*
>
> A5: As we argued before in A3, VQA-based neuro-symbolic frameworks assume that natural language sentences are given as input. However, Kandinsky patterns and CLEVR-Hans data set do not provide questions in natural language sentences for each visual input. Therefore, VQA-based neuro-symbolic frameworks cannot be applied directly. Ideally, we would like to address the task to learn the abstract patterns defined on the high-level concepts from visual inputs. NSFR is the first step to building such a framework.

---

> > ### Author Response · Authors · 2021-11-15
> > **Response to Reviewer hJBE (Part 2/3)**
> >
> > *Q6: What makes Kandisky patterns more challenging?*
> >
> > A6: The difficulty of the Kandinsky patterns is that the classification patterns are defined on the high-level concepts, using the relations of the objects and the attributes. In our experiments with relatively complex Kandinsky patterns such as red-triangle, online/pair, and 9-circles, the performance of the benchmark model without logic has degraded. This indicates that relational representations and reasoning are crucial to solving these problems. We would also argue here that the deficiency of standard pattern recognition techniques has been pinpointed by interdisciplinary scientists in the past several decades. Over fifty years ago, M. M. Bongard, a Russian computer scientist, invented a collection of one hundred human-designed visual recognition tasks, now named the Bongard Problems (BPs), to demonstrate the gap between high-level human cognition and computerized pattern recognition. With the same motivation, the Bongard-LOGO [10] problem has been proposed.  The Kandinsky patterns problem [11] is on the same line.
> >
> >
> > *Q7: “More recently, several neuro-symbolic techniques for commonsense reasoning (Arabshahi et al., 2021), visual question answering (Mao et al., 2019; Amizadeh et al., 2020) and multimedia tasks (Khan & Curry, 2020) have been developed. They either do not employ a differentiable forward reasoner or miss object-centric learning in the end-to-end reasoning architecture.” It would be helpful to be clear about which methods fall in to which category.*
> >
> > A7: None of these architectures use differentiable forward-chaining reasoning on first-order logic. Mao [3], Amizadeh [12], and Khan [13] have an object-centric perception module, and the other does not address the reasoning tasks on the objects in visual inputs.
> >
> >
> > *Q8: “NSFR encapsulates different object-perception models, thus allows us to choose a proper model depending on the situation and the problem to be solved.” Does this mean that NSFR can learn both with and without supervision? However, later in the paper it appears that you do need supervision? Please clarify.*
> >
> > A8: We apologize for being unclear. In our setting we need supervision and what we meant here is that we can employ different perception models.  For example, we can employ a slot attention module for the CLEVR dataset. However, with natural images, we can employ other established object-detection models such as YOLO [1] and Faster-RCNN [14]. In general, the object-detection model requires labels for the bounding boxes of the objects, which are not required to train the slot attention module. However, slot attention cannot handle natural images, which can be handled with an object-detection model. To manage the trade-offs, NSFR offers the capability to choose the proper model depending on the problem to be solved.
> >
> >
> > *Q9: exactly what the contributions of the paper are and what the related work is? The related work section is very hard to follow*
> >
> > A9: Thank you for pointing it out. We will improve the related work section, highlighting the difference from previous neuro-symbolic works. We appreciate the suggestion.
> >
> >
> > *Q10: “The output of the visual-perception module is already factorized in terms of objects.” How is this assumption enforced?*
> >
> > A10: The output of the visual-perception model should be decomposed for each object, as in Figure 2. For instance, a feature map for an entire input image is not acceptable as the output of the perception module.
> >
> >
> > *Q11: In figure 3 and Example 2 it is not clear what a “valuation function” is. What is defined only ...?*
> >
> > A11: A valuation function is defined on each neural predicate in the language. A valuation function for predicate p computes the probability over ground atoms which consist of predicate p. For instance, v_p computes probabilities for atoms {p(a,a), p(a,b), ...} given visual input.
> > The domain of the valuation function is determined by the data types of the predicate. In the object-centric reasoning language, which is defined in Def. 2, the set of constants can be divided in terms of data types.
> >
> >
> > *Q12: Where do the ground atoms in Figure 3 come from?? It appears that you can have significantly more ground atoms than are necessary to describe the two scene? Do you need to define new ground atom for each task? It would be helpful to discuss this.*
> >
> > A12: Yes, the ground atoms need to be defined for each task. In our experiments, we defined a language for Kandinsky patterns and CLEVR. Then we generated possible ground atoms automatically from the predicates and constants in the language. We enumerated the ground atoms based on the given language, i.e., we generated all of the ground atoms. The enumeration of the necessary and sufficient set of ground atoms for a given task has been addressed in [7]. NSFR can naturally adopt this enumeration approach.

---

> > > ### Author Response · Authors · 2021-11-15
> > > **Response to Reviewer hJBE (Part 3/3)**
> > >
> > > *Q13: Table 1: Why not compare to other Neuro-Symbolic methods such as NeuroSymbolic concept learner or NeuralSymbolic VQA? Or even using a relation net (A simple neural network module for relational reasoning, Santoro et al)?*
> > >
> > > A13: As we argued in A3, NSFR is not dedicated to VQA tasks. All of the VQA-based models assume that the input is a pair of an image and a question in the form of a natural language sentence. Kandinsky patterns and CLEVR-Hans data sets do not assume that each image is associated with a sentence. Therefore, VQA-based Neuro-Symbolic frameworks can not be applied directly.
> > >
> > >
> > > *Q14: Table 1 and 2: How many runs were done? What is the variance? It would be helpful to see the training curves to better understand the overfitting in table 1.*
> > >
> > > A14: Thank you for pointing it out. In the experiments, we performed one run for each data set, because the trainable modules in NSFR are all pre-trained. For the same reason, the result of the inference does not result in much variance with respect to the random seed.
> > >
> > >
> > > *Q15: What are the losses used for training? These are not described anywhere?*
> > >
> > > A15: We used the following loss functions for the training steps.
> > > - Perception Module:
> > >   - YOLO: We used the loss function that approximates detection performance, presented in [1].
> > >   - Slot Attention: We trained the slot attention model with the set prediction architecture following [2], using the loss function which is based on the Hungarian algorithm [15].
> > > - Neural Predicates (Concept Learning): We used the binary-cross entropy loss.
> > > - Benchmark models in the Kandinsky patterns data set: We used the binary-cross entropy loss.
> > >
> > >
> > > Finally, thank you for pointing out the typos. We will fix them.
> > >
> > >
> > > [1] Joseph Redmon, Santosh Divvala, Ross Girshick, Ali Farhadi: You Only Look Once: Unified, Real-Time Object Detection, CVPR, 2016.
> > >
> > > [2] Francesco Locatello, Dirk Weissenborn, Thomas Unterthiner, Aravindh Mahendran, Georg Heigold, Jakob Uszkoreit, Alexey Dosovitskiy, Thomas Kipf: Object-Centric Learning with Slot Attention, NeurIPS, 2020.
> > >
> > > [3] Jiayuan Mao, Chuang Gan, Pushmeet Kohli, Joshua B. Tenenbaum,Jiajun Wu: The Neuro-Symbolic Concept Learner: Interpreting Scenes, Words, and Sentences From Natural Supervision, ICLR, 2019.
> > >
> > > [4] Stephen Muggleton.: Inductive logic programming. New Generation Computing 8(4): 295–318, 1991.
> > >
> > > [5] Andrew Cropper, Sebastijan Dumančić, Stephen H. Muggleton: Turning 30: New Ideas in Inductive Logic Programming, IJCAI, 2020.
> > >
> > > [6] Richard Evans, Edward Grefenstette: Learning Explanatory Rules from Noisy Data, JAIR, 2018.
> > >
> > > [7] Hikaru Shindo, Masaaki Nishino, Akihiro Yamamoto.: Differentiable Inductive Logic Programming for Structured Examples. AAAI, 2021.
> > >
> > > [8] Robin Manhaeve, Sebastijan Dumančić, Angelika Kimmig, Thomas Demeester, Luc De Raedt: DeepProbLog: Neural Probabilistic Logic Programming, NeurIPS, 2018.
> > >
> > > [9] Zhun Yang, Adam Ishay, Joohyung Lee: NeurASP: Embracing Neural Networks into Answer Set Programming, IJCAI, 2020.
> > >
> > > [10] Weili Nie, Zhiding Yu, Lei Mao, Ankit B. Patel, Yuke Zhu, Anima Anandkumar: Bongard-LOGO: A New Benchmark for Human-Level Concept Learning and Reasoning, NeurIPS, 2020.
> > >
> > > [11] Heimo Müller, Andreas Holzinger: Kandinsky Patterns, Artificial Intelligence, Volume 300, 103546, 2021.
> > >
> > > [12]  Saeed Amizadeh, Hamid Palangi, Oleksandr Polozov, Yichen Huang, Kazuhito Koishida: Neuro-Symbolic Visual Reasoning: Disentangling "Visual" from "Reasoning", ICML, 2020.
> > >
> > > [13] Muhammad Jaleed Khan, Edward Curry: Neuro-symbolic Visual Reasoning for Multimedia Event Processing: Overview, Prospects and Challenges, ICKM (Workshop), 2020.
> > >
> > > [14] Shaoqing Ren, Kaiming He, Ross Girshick, Jian Sun: Faster R-CNN: Towards Real-Time Object Detection with Region Proposal Networks, NIPS, 2015.
> > >
> > > [15] Harold W Kuhn. The Hungarian method for the assignment problem. Naval research logistics quarterly, 2 (1-2):83–97, 1955.

---

> > > > ### Comment · Reviewer_hJBE · 2021-11-18
> > > > **Interesting direction, but still some work is needed for it to be suitable for ICLR.**
> > > >
> > > > Overview:
> > > > 1. The authors claim that the paper is not “about learning”,  and the parts of the paper that do involve learning are not novel. Therefore, it is not clear that this paper is appropriate for ICLR.
> > > > 2. The authors claim that their work is the “first step to establishing the framework to combine the ILP with object-centric visual input”. While this is certainly exciting, the authors have not made any novel technical or empirical contributions, rather than have combined existing architectures. This paper appears to be a work in progress and could be improved by demonstrating ILP results. This work is certainly worth sharing in a workshop format, but is currently not suitable for ICLR.
> > > >
> > > >
> > > > Q1A1(iii): Proving the set of rules required for reasoning is a weakness of the paper. There are other papers such as NeuroSymbolic concept learner that learn to compose functions in order to solve reasoning problems.
> > > >
> > > > Q3A3(i): “symbolic programs are determined by the natural language sentences but NSFR does not have the assumption” - Why is this a disadvantage?
> > > >
> > > > Q3A3(i): “We believe that NSFR is the first step to establishing the framework to combine the ILP with object-centric visual input.” - This would certainly be interesting. Can the authors provide any preliminary results in this direction? That would strengthen the paper.
> > > >
> > > > Q8A8: Thanks for clarifying.
> > > >
> > > > Q15A15: Please update the paper to include the losses.

---

> > > > > ### Author Response · Authors · 2021-11-19
> > > > > **Response to Reviewer hJBE**
> > > > >
> > > > > Thanks for the response. We answer the raised questions pointwise.
> > > > >
> > > > >
> > > > > *Overview*
> > > > >
> > > > > A1: We believe that reasoning is an essential base of learning, and developing reasoning helps develop learning, specifically in neuro-symbolic approaches. We would note here that the field of probabilistic logic programming [16] (a.k.a. Statistical Relational Learning [17]), which aims to integrate probabilities with symbolic logic, has been developed on many works whose focus is reasoning. Therefore, we believe that reasoning and learning can be discussed separately, with an assumption that learning algorithms are developed on the reasoning architecture.
> > > > >
> > > > >
> > > > > *Q1A1(iii): Proving the set of rules required for reasoning is a weakness of the paper. There are other papers such as NeuroSymbolic concept learner that learn to compose functions in order to solve reasoning problems.*
> > > > >
> > > > > A2: Neuro-Symbolic Concept Learner learns how to generate programs from natural language sentences, i.e., learns how to parse the question to the symbolic program. We think that this is a different problem setting from *inductive learning* of symbolic programs from examples, which NSFR aims.
> > > > >
> > > > >
> > > > > *Q3A3(i): “symbolic programs are determined by the natural language sentences but NSFR does not have the assumption” - Why is this a disadvantage?*
> > > > >
> > > > > A3: The assumption causes problems if users address inductive programming of abstract patterns. Ideally, Kandinsky patterns and CLEVR-Hans data set requires the models to explain the abstract patterns of visual inputs, i.e., learning explicit rules from visual data. Thus, in this setting, models should perform symbolic induction only from visual inputs. The models that need questions in natural language sentences are not directly applicable.
> > > > >
> > > > >
> > > > > *Q3A3(i): “We believe that NSFR is the first step to establishing the framework to combine the ILP with object-centric visual input.” - This would certainly be interesting. Can the authors provide any preliminary results in this direction? That would strengthen the paper.*
> > > > >
> > > > > A4: We did not yet perform structure learning, i.e., learning explicit logical rules from data. The reason is that the evaluation of the structure learning is not easy because of the *language biases*. This problem has been discussed in [18]. In essence, in the rule learners, users need to provide the language biases, to determine the search space. For example, users can give a template that specifies a set of rules to be generated. If a user can provide suitable biases, then the rule learner is very efficient. However, if a user cannot provide suitable biases, then the rule learner is almost useless. The same arguments can be applied to structure learning in NSFR.  It is not trivial how to decide a proper language bias for the object-centric reasoning problems. More importantly, it is not clear how to compare fairly with deep learning models. This is an open question in neuro-symbolic researches that involve structure learning.
> > > > > We would note here that our formulation allows weights for rules. If a proper language bias is given, then the rules can be learned by gradient descent in NSFR because the problem of rule learning by gradient descent has been addressed in [7], and NSFR takes the same approach in the reasoning process.
> > > > > We would also note here that even the differentiable rule learner [6], which achieves state-of-the-art results in many tasks on the symbolic domain, do not learn complex rules to describe the scenes for Kandinsky patterns and CLEVR Hans data set, because of the memory-consumption problem. If we allow the complex rules to describe the scenes, the differentiable rule learner is going to run out of memory. For example, in the red-triangle data set, NSFR takes a rule that contains a constant *red* in the body. This type of rule is not allowed in the differentiable rule learner.
> > > > >
> > > > > *Q15A15: Please update the paper to include the losses.*
> > > > >
> > > > > A5: We will include the descriptions of the loss functions.
> > > > >
> > > > >
> > > > > [16] Daan Fierens, Guy Van den Broeck, Joris Renkens, Dimitar Shterionov, Bernd Gutmann, Ingo Thon, Gerda Janssens, and Luc De Raed: Inference and learning in probabilistic logic programs using weighted Boolean formulas, Theory and Practice of Logic Programming, 2015.
> > > > >
> > > > > [17] Luc De Raedt, Kristian Kersting, Sriraam Natarajan, David Poole: Statistical Relational Artificial Intelligence: Logic, Probability, and Computation. Synthesis Lectures on Artificial Intelligence and Machine Learning. Morgan & Claypool Publishers, 2016.
> > > > >
> > > > > [18] Andrew Cropper, Sebastijan Dumančić, Richard Evans, Stephen H. Muggleton: Inductive logic programming at 30, Machine Learning, 2021.

---

### Author Response · Authors · 2021-11-15
**General Comment for all Reviewers**

We would like to point out various misconceptions that exist in the different reviews.
1. There seems to be a common misconception that our work is about learning. We would like to clarify that our work focuses on **reasoning**, which is a separate problem. Even neuro-symbolic methods, such as Neuro-Symbolic concept learner (Mao et al. ICLR 2019), do not perform explicit structure learning with respect to abstract patterns. A simple form of learning can be implemented with reasoning. For example, we can train the weights for rules or work with neural networks, e.g., set transformers (or similar architectures), to select relevant rules, but we leave that for a separate paper.

2. Another misconception the reviewers have is that since we propose a neuro-symbolic method, experimenting on visual question answering tasks is necessary. We would like to point out that our aim is not solving VQA tasks but establishing a logic-based neuro-symbolic framework that can perform reasoning on visual inputs and be incorporated with symbolic inductive learning techniques.

---

> ### Comment · Reviewer_Jwkf · 2021-11-22
> **Response to general comment to all reviewers**
>
> The authors write:
>
> "There seems to be a common misconception that our work is about learning."
>
> I suspect the other reviewers, like myself, felt that the paper *ought* to be about learning, since ICLR is a learning conference. Perhaps the paper would be better suited to a general AI conference, such as IJCAI, AAAI, or ECAI.

---

> > ### Author Response · Authors · 2021-11-28
> > **This is a partisan comment !!**
> >
> > It seems that the reviewer has a pre-conceived notion that all ICLR papers **ought** to be about learning. This is frustrating and amusing at the same time to see at a top ML conference.
> > If this was the case then papers such as:
> > 1. Differentiable Reasoning over a Virtual Knowledge Base, Dhingra et al., ICLR 2020 (https://openreview.net/pdf?id=SJxstlHFPH)
> > 2. Transformer-XH: Multi-Evidence Reasoning with eXtra Hop Attention, Zhao et al, ICLR 2020 (https://openreview.net/pdf?id=r1eIiCNYwS)
> > 3. Neural Module Networks for Reasoning over Text, Gupta et al, ICLR 2020 (https://openreview.net/pdf?id=SygWvAVFPr)
> > 4. Hopper: Multi-hop Transformer for Spatiotemporal Reasoning, Zhou et al., ICLR 2021 (https://openreview.net/pdf?id=MaZFq7bJif7)
> >
> > would not have been accepted to ICLR.

---

> > > ### Comment · Reviewer_Jwkf · 2021-11-28
> > > **Reply to authors**
> > >
> > > I am more than happy to defer to the AC and / or other reviewers on this issue if the consensus is that ICLR can take papers that are not about learning. However, I note that all four of the papers you cite certainly are about learning (as well as other things, including reasoning). I mean no disrespect to your work - there is certainly more to AI than learning - so please don't take offence at the comment. But ICLR is a learning conference (and, historically at least, primarily a deep learning conference; see the call for papers). So it's understandable that reviewers are chiefly looking for a contribution to the learning field.

---

### Decision · Program_Chairs · 2022-01-20

**Decision:**

Reject

**Comment:**

Unfortunately, the reviewers have unanimously voted to reject this paper.
There was some discussion of whether the paper was out-of-scope for ICLR;
I don't think that it is, necessarily, but I think that we can kind of screen off that topic because the reviewers had plenty of non-scope-related concerns that seem disqualifying to me, including both issues of novelty and issues related to the experimental validation.
Therefore, I am also recommending rejection in this case.